# Transition between fermentation and respiration determines history-dependent behavior in fluctuating carbon sources

**Bram Cerulus[1,2†], Abbas Jariani[1,2†], Gemma Perez-Samper[1,2], Lieselotte Vermeersch[1,2], Julian MJ Pietsch[3], Matthew M Crane[3,4], Aaron M New[1,2], Brigida Gallone[1,2,5,6], Miguel Roncoroni[1,2], Maria C Dzialo[1,2], Sander K Govers[1,2], Jhana O Hendrickx[1,2], Eva Galle[1,2], Maarten Coomans[1,2], Pieter Berden[1,2], Sara Verbandt[1,2], Peter S Swain[3], Kevin J Verstrepen[1,2]\***

[1]VIB Laboratory for Systems Biology, VIB-KU Leuven Center for Microbiology, Leuven, Belgium; [2]Departement Microbiële en Moleculaire Systemen (M2S), CMPG Laboratory of Genetics and Genomics, Leuven, Belgium; [3]Centre for Synthetic and Systems Biology, School of Biological Sciences, University of Edinburgh, Edinburgh, United Kingdom; [4]Department of Pathology, University of Washington, Washington, United States; [5]Department of Plant Biotechnology and Bioinformatics, Ghent University, Ghent, Belgium; [6]VIB Center for Plant Systems Biology, Ghent, Belgium

**Abstract** Cells constantly adapt to environmental fluctuations. These physiological changes require time and therefore cause a lag phase during which the cells do not function optimally. Interestingly, past exposure to an environmental condition can shorten the time needed to adapt when the condition re-occurs, even in daughter cells that never directly encountered the initial condition. Here, we use the molecular toolbox of *Saccharomyces cerevisiae* to systematically unravel the molecular mechanism underlying such history-dependent behavior in transitions between glucose and maltose. In contrast to previous hypotheses, the behavior does not depend on persistence of proteins involved in metabolism of a specific sugar. Instead, presence of glucose induces a gradual decline in the cells' ability to activate respiration, which is needed to metabolize alternative carbon sources. These results reveal how trans-generational transitions in central carbon metabolism generate history-dependent behavior in yeast, and provide a mechanistic framework for similar phenomena in other cell types.
DOI: https://doi.org/10.7554/eLife.39234.001

**\*For correspondence:**
kevin.verstrepen@kuleuven.vib.be

[†]These authors contributed equally to this work

**Competing interests:** The authors declare that no competing interests exist.

## Introduction

Cells can face sudden and dramatic changes in their environment such as nutrient depletion, temperature shifts, and osmotic shock. In general, cells adapt to such changes by activating and repressing specific genes and processes required to function in the new environment (*Brewster et al., 1993*; *Gasch et al., 2000*; *Jacob and Monod, 1961*; *Mitchell et al., 2009a*; *Perez-Samper et al., 2018*). Such physiological re-programming can take a considerable amount of time and resources during which the cells do not function optimally, a phenomenon known as the lag phase. Interestingly, recent exposure to a specific environment can shorten the lag time needed to adapt when the same stimulus re-occurs, even in daughter cells that have not directly experienced the initial change (*D'Urso et al., 2016*; *Guan et al., 2012*; *Zacharioudakis et al., 2007*). This phenomenon is referred to as 'history-dependent behavior' (HDB). HDB is believed to allow cells to more quickly adapt (i.e. reduced lag duration) when a similar environment returns. Such HDB often extends over several

cellular generations, and is therefore sometimes considered to be a form of rudimentary epigenetic behavior or 'cellular memory'.

HDB has been described in various organisms and cell types, ranging from microbes to humans. Some of the best-documented cases, sometimes referred to as 'metabolic programming', involve a switch in nutrient status. One example occurs in pancreatic cells, where previous exposure to high glucose levels leads to long-lasting stress and damage, even after the glucose levels have been normalized, and inversely, prolonged tight control of glycemic values results in lasting improvements, even when glucose levels are not tightly controlled (*Tonna et al., 2010*). Similarly, prolonged changes in blood glucose levels linked to dietary changes may affect muscle and adipose tissue development, even in a next generation that has never been directly exposed to the nutritional stress (*Sharples et al., 2016*). Furthermore, trans-generational shift in metabolic state is also observed in exponentially growing yeast cells (*Slavov et al., 2014*). The mechanisms underlying metabolic programming have not been fully elucidated, although changes in chromatin state have been implicated in the effect.

HDB is easily studied in single-cell organisms because during the lag phase, these cells arrest their growth, making it easy to observe the adaptation process (*Acar et al., 2008*; *Brickner, 2010*; *D'Urso et al., 2016*; *Friedman et al., 2014*; *Guan et al., 2012*; *Kundu and Peterson, 2010*; *Light et al., 2010*; *Mitchell et al., 2009b*; *Stockwell et al., 2015*). One of the best-documented examples of HDB occurs when *Saccharomyces cerevisiae* cells are repeatedly shifted between glucose and galactose (*Stockwell et al., 2015*). The first shift from glucose to galactose generates a slow induction of the *GAL* genes, with an associated long lag phase. When the same population is returned to glucose and subsequently switched back to galactose, the *GAL* induction rate and growth response is significantly faster. This HDB can extend for up to 12 hr after the shift from galactose to glucose. The 12 h-period in glucose during which the HDB is retained corresponds to approximately five cellular generations, at which point less than 4% of the cells has directly experienced galactose before (*Kundu and Peterson, 2010*; *Sood et al., 2017*; *Stockwell and Rifkin, 2017*; *Stockwell et al., 2015*; *Zacharioudakis et al., 2007*). A similar phenomenon occurs when *S. cerevisiae* cells are switched between glucose and maltose (*New et al., 2014*), and when *E. coli* cells are switched between glucose and lactose (*Lambert et al., 2014*).

The molecular principles underlying this type of HDB are only recently being uncovered. In general, transcriptional induction of genes which are crucial for rapid growth in the inducing environment (e.g. *GAL* gene induction in galactose) are assumed to be the rate-limiting step determining the length of the lag phase (*Lambert et al., 2014*; *New et al., 2014*; *Wang et al., 2015*). As a consequence, HDB observed at the level of growth is often thought to be linked to a similar effect in the induction of specific genes. More specifically, the regulatory networks governing induction of these specific genes are believed to have intrinsic properties that allow faster re-induction if the genes have been recently induced, which in turn leads to a faster resumption of cellular growth (*D'Urso et al., 2016*; *Stockwell et al., 2015*; *Zacharioudakis et al., 2007*). Importantly, however, the assumption that growth resumption is directly governed by the induction kinetics of nutrient-specific genes has not been supported by strong experimental evidence.

Two major molecular mechanisms have been proposed for HDB on the level of transcription. First, a previous induction of a gene may generate an epigenetically heritable shift in local chromatin structure that allows for quicker re-induction after a short time in the repressive condition (*Brickner, 2010*; *Brickner et al., 2007*; *D'Urso et al., 2016*; *Tan-Wong et al., 2009*). The second proposed mechanism is the transgenerational persistence of specific proteins, referred to as 'protein inheritance' or 'protein perdurance'. This mechanism assumes that proteins needed in one environment do not immediately disappear when cells are shifted to a new environment. During cell division, some of these lingering proteins can be transmitted to the daughter cell and influence how this cell functions, leading to HDB. One of the best-studied examples of such protein inheritance occurs in galactose-to-glucose shifts in *S. cerevisiae*. Cells growing in galactose have high levels of the galactokinase enzyme Gal1p. When these cells are shifted to glucose, the *GAL1* gene is repressed and the Gal1 proteins that were present are gradually diluted as the cells divide. However, when the cells are shifted again to galactose before the cellular Gal1p levels reach a basal level, the remaining Gal1 proteins may allow faster re-induction of the whole set of Gal proteins needed to resume growth (*Stockwell et al., 2015*; *Zacharioudakis et al., 2007*).

Though these are both plausible mechanisms, it is important to note that the causative role of Gal1 protein inheritance has not yet been confirmed and that environmental shifts can cause much broader changes in transcriptional activity. Transitions to glucose stimulate genes involved in fermentation and cellular growth and repress genes involved in respiration, the glyoxylate and TCA cycles, gluconeogenesis, synthesis of fatty acids and storage carbohydrates, and stress resistance (*Conrad et al., 2014*; *Zaman et al., 2008*). Some of these processes could be prime determinants (bottlenecks) of the lag time, while the activation of the *GAL* genes is a vital last step to resume growth, but could be less important in determining the lag time and HDB.

To obtain a systematic and thorough understanding of HDB in fluctuating carbon sources, we combine genome-wide assays (Bar-Seq and transcriptomics) with various single-cell analyses. Specifically, we show that the duration of the lag phase when cells are switched from glucose to maltose depends on the duration of growth in glucose, with longer exposure to glucose eliciting longer lag times. In contrast to what has been previously suggested, our results show that this HDB does not rely on inheritance of dedicated nutrient-specific proteins (in this case the Mal proteins). Instead, several lines of evidence point towards the existence of a much broader mechanism for HDB than previously assumed. Repression of respiratory pathways during glucose growth and the subsequent re-induction upon shift to maltose appears to be playing a key role in HDB. Firstly, we observe that not only pre-growth in maltose, but also pre-growth in galactose leads to shorter lag in maltose. Secondly, using heterokaryon cells with two nuclei with distinct pre-growth conditions, we show that the HDB is linked to the cytoplasm rather than the nucleus. Thirdly, genome-wide assays show that genes involved in mitochondrial function and respiration play a central role in HDB. Fourthly, inhibiting respiration prolongs the lag phase even in cells that only grew for short periods in glucose. Finally, over-activating respiration through overexpression of the *HAP4* respiratory regulator abolishes the HDB and results in nearly constant short lag phases.

Taken together, our data shows that, in contrast to previous suggestions, HDB in changing carbon environments is not the consequence of changes in chromatin state or persistence of a few metabolism-specific proteins. Instead, the effect depends on a much broader metabolic network, with slow changes in respiration and fermentation playing a key role.

## Results

### *S. cerevisiae* demonstrates strong HDB for glucose-to-maltose and glucose-to-galactose shifts

Cells demonstrate HDB for the shift between glucose and maltose (*Figure 1* and *Figure 1—figure supplement 1*). Cultures of *S. cerevisiae* were adapted to maltose and subsequently transferred to glucose for different lengths of time (0, 2, 4, 6, 8 and 24 hr). The cells were then transferred back to maltose, where they experienced a lag phase before resuming growth (*Figure 1A*). The duration of the respective lag phases was measured both at the population level and at the single-cell level (*Figure 1B–E*). Both methods establish that the duration of the lag phase depends greatly on the duration of the preceding glucose growth, with longer glucose exposure leading to longer lag times. Interestingly, our single-cell measurements show that the increasing lag times are a result of individual cells taking more time to resume growth, as well as an increasing fraction of cells that do not resume growth at all (*Figure 1E*). Moreover, this HDB is not unique to maltose-glucose-maltose shifts. We observe a similar behavior in galactose-glucose-galactose shifts and also in glycerol-glucose-glycerol shifts, suggesting that this is a commonly occurring phenomenon (*Figure 1—figure supplements 1* and *2*).

Although we always kept the cell populations very dilute, we wanted to confirm that HDB is not due to alterations in nutrient content of media during extended growth times. We therefore measured single cell lag times after glucose-glucose-glucose shifts, with different durations of intermediate glucose growth. No detectable pattern in the dynamics of growth resumption upon shift to the new media is observed (*Figure 1—figure supplement 3*).

Finally, we observe that HDB is also present, albeit to different extents, in L-1374 and BC187, two non-laboratory strains (*Cubillos et al., 2009*) (*Figure 1—figure supplement 4*).

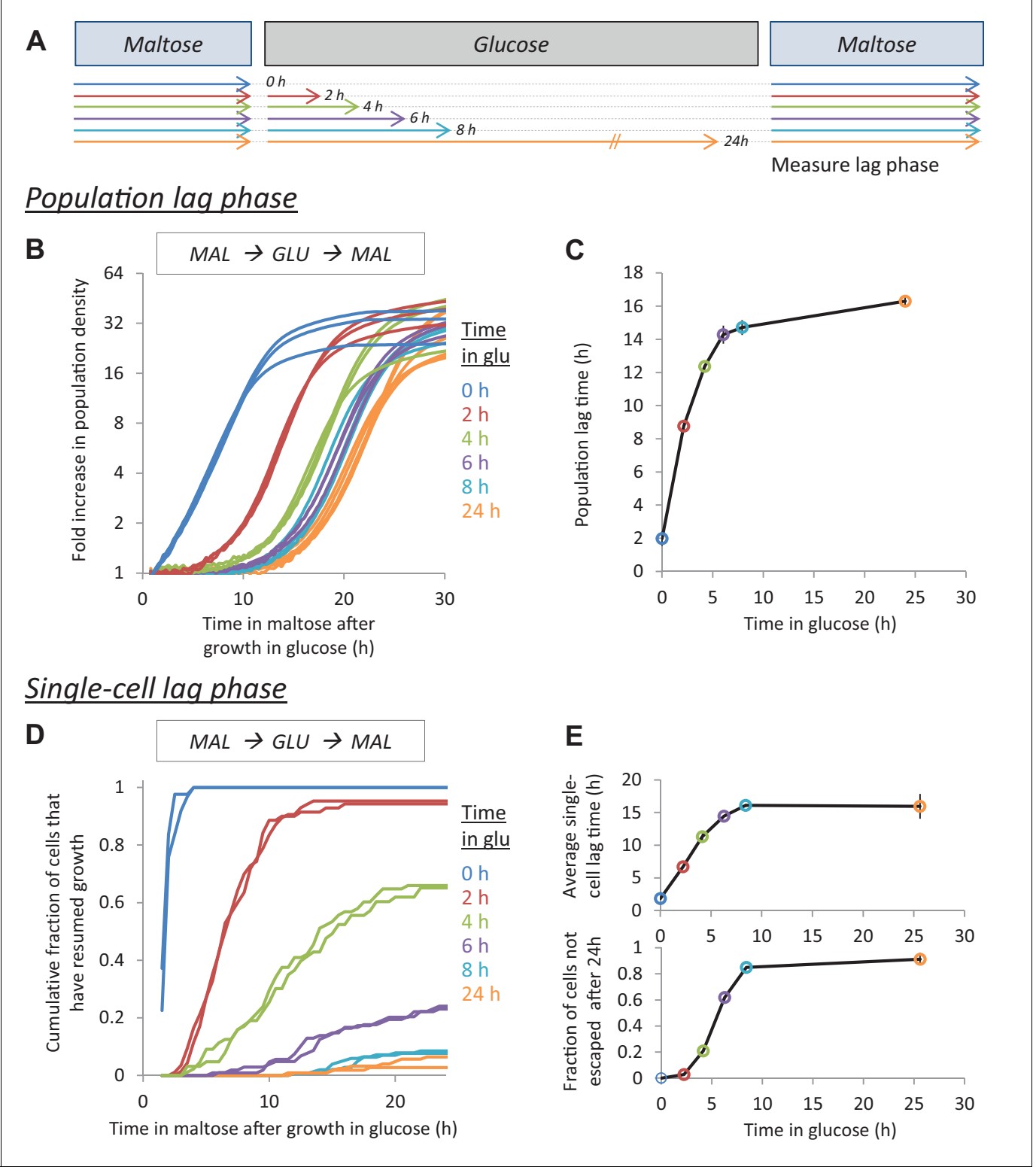

**Figure 1.** The lag time after glucose-maltose shifts depends on the time grown in glucose. (A) Experimental set-up for measuring HDB. Cultures adapted to maltose growth are transferred to glucose for different times (0, 2, 4, 6, 8, 24 hr). Then, these cultures are transferred back to maltose where they experience a lag phase. The lag time is measured either on the population-level (B,C) or on the single-cell level (D,E). (B) Population lag times are measured by tracking population density ($OD_{600}$) after the shift to maltose. The lag time is manifested by the delayed increase in cell density after glucose periods. (C) Quantification of the population lag time for the data shown in (B). (D) Single-cell lag times are measured using time-lapse microscopy. The cumulative fraction of cells initially present after the shift that have resumed growth is plotted against time in maltose. (E) Summary

*Figure 1 continued on next page*

*Figure 1 continued*

statistics of the single-cell lag time data shown in (D). We summarize these data by calculating (upper) the mean lag time of the cells resuming growth, (lower) the fraction of cells that did not resume growth before the end of the experiment. The error bars in (C) and (E) represent the range for two replicates.

DOI: https://doi.org/10.7554/eLife.39234.002

The following figure supplements are available for figure 1:

**Figure supplement 1.** The lag time after shifts to galactose depends on the time grown in glucose.

DOI: https://doi.org/10.7554/eLife.39234.003

**Figure supplement 2.** The lag time after shifts to glycerol depends on the time grown in glucose Accumulative distribution of single-cell lag times for glucose-glycerol shifts after pre-growth in glycerol.

DOI: https://doi.org/10.7554/eLife.39234.004

**Figure supplement 3.** Different periods of growth on glucose does not alter growth resumption dynamics after shift to fresh glucose.

DOI: https://doi.org/10.7554/eLife.39234.005

**Figure supplement 4.** HDB in non-laboratory strains.

DOI: https://doi.org/10.7554/eLife.39234.006

**Figure supplement 5.** The effect of cell density before the shift to glucose on lag time.

DOI: https://doi.org/10.7554/eLife.39234.007

**Figure supplement 6.** The effect of cell density before the shift to maltose on lag time.

DOI: https://doi.org/10.7554/eLife.39234.008

## HDB does not depend on mal or gal protein inheritance

Initially, we hypothesized that HDB after a shift to maltose might be explained by the same molecular mechanisms that have been suggested to underlie HDB after a shift from glucose to galactose, where perdurance of Gal proteins after a shift from galactose to glucose has been suggested to shorten the lag phase when the cells are again shifted to galactose. Thus, we hypothesized that proteins required for maltose metabolism could persist for some time during growth on glucose and might enable swift restart of growth when maltose is present again (*Zacharioudakis et al., 2007*).

A simple way to investigate whether Mal protein perdurance underlies the HDB is to change the pre-growth on maltose to a pre-growth on galactose. In galactose, the intracellular concentration of the maltose-cleaving enzyme, Mal12p, measured using fluorescent protein fusions, was approximately 70 times less than its concentration in maltose, and we were unable to detect the presence of the maltose-transporter Mal1p (*Figure 2C*). We thus compared the lag duration in maltose-glucose-maltose shifts to galactose-glucose-maltose shifts (*Figure 2A* and *Figure 2—figure supplement 1*). Surprisingly, compared to pre-growth in maltose, pre-growth in galactose actually increased the timescale over which HDB occurs and generally reduced the lag time after short glucose periods. This is the opposite of what is expected if the inheritance of Mal proteins would be solely responsible for the HDB after glucose-to-maltose shifts.

Although the Mal proteins are gradually diluted over several generations after a switch from maltose to glucose (*Figure 2C*), the persistence of the Mal proteins does not cause a shorter lag phase after the shift back into maltose. Indeed, even though the correlation between Mal12p/Mal11p levels and the population lag time is strong after maltose pre-growth, there is little correlation between Mal protein levels and glucose-to-maltose lag duration when cells are first pre-grown in galactose before they are shifted to glucose and then to maltose (*Figure 2E*). Therefore, the observed correlation after pre-growth in maltose seems to only reflect the number of generations of growth in glucose.

A similar set of experiments was performed to test whether Gal protein inheritance causes HDB after a glucose-galactose shift. Galactose pre-growth increased the timescale over which HDB occurs and resulted in generally shorter lags compared to the maltose pre-growth (*Figure 2B* and *Figure 2—figure supplement 2*). The inverse correlation between remaining Gal1p levels and lag times appears to support the previously reported hypothesis that Gal protein inheritance is indeed responsible for quicker re-activation of *GAL* genes and subsequent escape from the lag phase (*Zacharioudakis et al., 2007*). However, pre-growth on maltose, which does not result in measurable Gal1p levels (*Figure 2D*), also leads to considerable HDB (*Figure 2B*). Moreover, whereas there is a clear correlation between Gal1p levels and the population lag time after galactose pre-growth, there is little correlation when cells are pre-grown on maltose (*Figure 2F*). Together, these results suggest

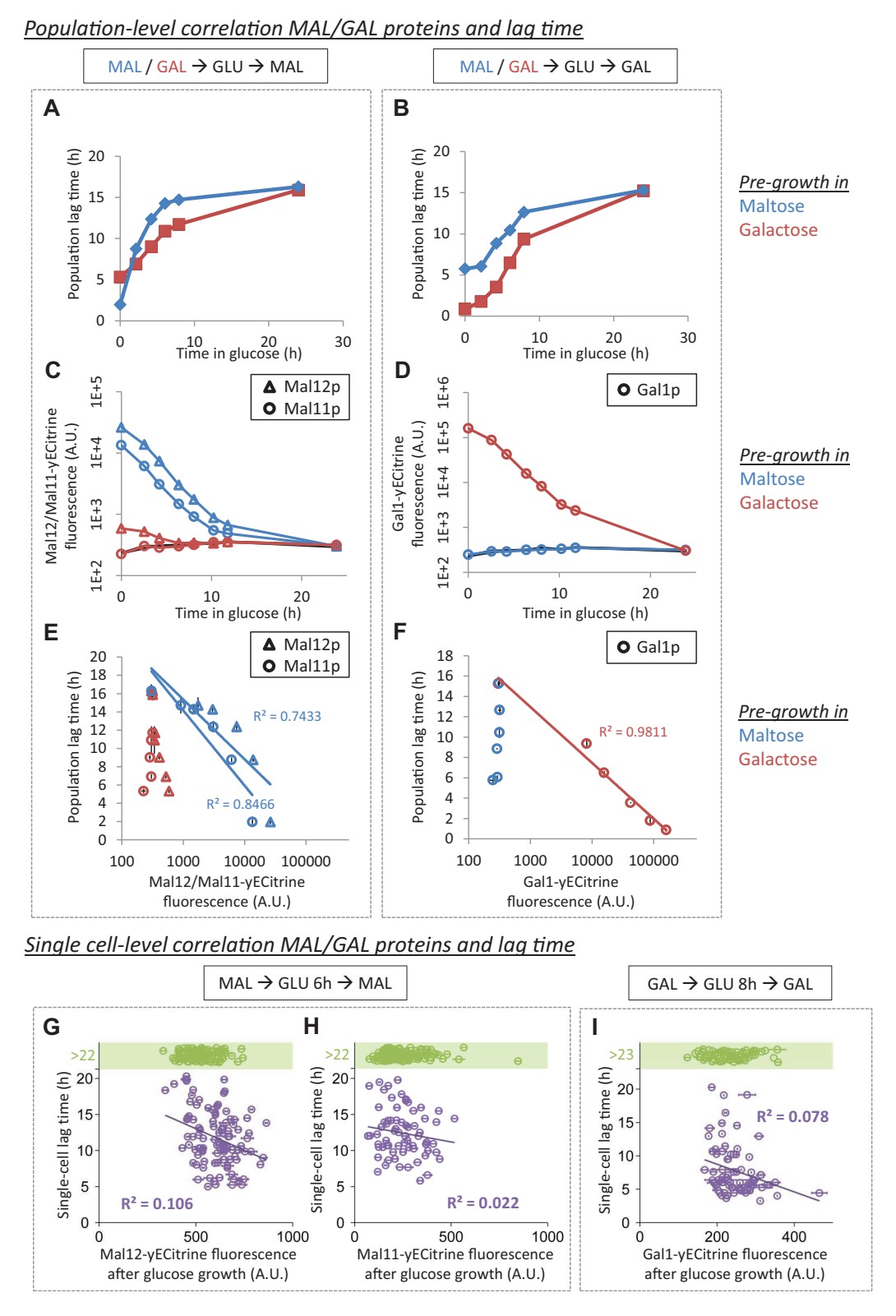

**Figure 2.** *MAL/GAL protein levels are not always correlated with the lag time.* (**A**) The effect of pre-growth in maltose or galactose on HDB after glucose-maltose shifts. (**B**) The effect of pre-growth in maltose or galactose on HDB after glucose-galactose shifts. (**C**) Inheritance of *MAL* proteins during glucose after pre-growth in maltose or galactose. (**D**) Inheritance of *GAL* proteins during glucose after pre-growth in maltose or galactose. (**E**) Correlation between lag time and *MAL* protein level after pre-growth in maltose or galactose for glucose-maltose shifts. (**F**) Correlation between lag

*Figure 2 continued on next page*

*Figure 2 continued*

time and *GAL* protein level after pre-growth in maltose or galactose for glucose-galactose shifts. The error bars for lag time measurement in (A), (B), (E) and (F) represent SEM for three replicates. The error bars for fluorescence measurement in (C) to (F) represent the range for two replicates. (G) Single-cell correlation for lag time and Mal12 protein level after maltose pre-growth, followed by 6 hr of glucose growth. The graph represents the mean and SD of fluorescence measurements taken between 2 hr and 3 hr after the shift to maltose. (H) Same as (G), but for Mal11. (I) Single-cell correlation for lag time and Gal1 protein level inherited after galactose pre-growth, followed by 8 hr of glucose growth. The graph represents the mean and SD of fluorescence measurements taken between 2 hr and 3 hr after the shift to galactose.

DOI: https://doi.org/10.7554/eLife.39234.009

The following figure supplements are available for figure 2:

**Figure supplement 1.** Single-cell lag times for glucose-maltose shifts with pre-growth in maltose or galactose.

DOI: https://doi.org/10.7554/eLife.39234.010

**Figure supplement 2.** Single-cell lag times for glucose-galactose shifts with pre-growth in maltose or galactose.

DOI: https://doi.org/10.7554/eLife.39234.011

**Figure supplement 3.** The effect of having or not having a bud on lag time.

DOI: https://doi.org/10.7554/eLife.39234.012

**Figure supplement 4.** The effect of replicative age on the single-cell lag time.

DOI: https://doi.org/10.7554/eLife.39234.013

that, similar to what we observed for the role of Mal proteins in shifts from glucose to maltose, Gal protein levels are not the major determinants of HDB after glucose-galactose shifts.

## Mal or gal protein inheritance does not explain cell-to-cell variability in lag duration

The lag duration of cells within an isogenic population is typically heterogeneous when observed at the single-cell level, and the heterogeneity increases with longer pre-growth in glucose before the switch to maltose or galactose. Some cells escape the lag phase within 6 hr, others take more than 20 hr, and some never resume growth within the timeframe of our experiments (*Figure 1D,E* and *Figure 1—figure supplement 1*). To investigate whether this variability is linked to protein perdurance, we correlated Mal and Gal protein expression levels immediately after cells leave glucose with the individual lag times after the respective carbon source shift (*Figure 2G,H,I*). In both cases, we find that variability in the level of inherited proteins does not explain the variability in single-cell lag times ($R^2 < 0.11$). Such absence of a strong correlation further confirms that there are other factors apart from Mal or Gal protein inheritance influencing the lag time after shifts from glucose to either maltose or galactose.

We hypothesized that the observed heterogeneity in single-cell lag times could potentially be explained by cell cycle stage or replicative age. We investigated the effect of cell cycle stage on lag time by observing whether the cells were budding at the time of the shift (*Figure 2—figure supplement 3*). This parameter was included as a covariate in a Cox proportional hazard model and was found to be a significant factor in 5 out of 16 experiments, increasing the likelihood of escape on average by 60%. However, this factor on average describes less than 4% of the variability in single-cell lag times (all Cox pseudo-$R^2 < 0.094$), indicating that the influence of cell cycle stage on single-cell lag times is neglectable. The effect of replicative age was investigated by staining bud scars with Calcofluor White (*Figure 2—figure supplement 4*). The number of bud scars was counted for individual cells, and this number was included as a covariate in the Cox model, together with the (un)budding covariate. Under this model, cells with different replicative ages were not found to have significantly different lag times.

In summary, the variability in single-cell lag times can neither be explained by variability in Mal and Gal protein levels nor by variability in cell cycle stage nor in replicative age.

## *MAL* gene induction correlates with escape from the glucose-to-maltose lag phase

Our results thus far indicate that HDB in shifts from glucose to maltose or galactose does not depend on inheritance of Mal or Gal proteins (*Figure 2*). However, this does not exclude the possibility that *MAL* or *GAL* gene induction is the final event that controls growth resumption. If this is the case, we would expect a strong correlation between the *MAL* or *GAL* gene induction time and

growth resumption within single cells. To test this, we used fluorescence time-lapse microscopy to track growth and gene expression within microcolonies during the lag phase (*Figure 3A*).

*MAL* gene induction dynamics after a glucose-to-maltose shift coincide with the resumption of rapid growth (*Figure 3B*). Within all observed microcolonies, *MAL* gene induction is followed by rapid and exponential growth and microcolonies that do not induce the *MAL* genes show little to no growth. Unexpectedly, we find many cells in which *MAL* gene induction is preceded by a period of slow growth (*Figure 3B*). However, the amount of growth without *MAL* gene induction is limited, as most microcolonies do not increase more than 2-fold in area before induction is detected (*Figure 3— figure supplements 1* and *2*).

It is possible that the slow growth indicates a period where *MAL* gene induction has started but proceeds at a slow rate. Slow induction could lead to production of low levels of Mal proteins that are undetectable by our methods, but are sufficient to allow growth. To investigate this, we measured the growth dynamics in a wild-type strain and a strain in which all *MAL* genes were deleted (*Figure 3D*). In general, the *mal* deletion mutant cannot grow well in maltose. Therefore, we pre-grew both strains in galactose before the shift. After the shift to maltose, the *mal* deletion strain is able to enter the slow growth phase but, unlike the wild-type, it is unable to shift to rapid exponential growth (*Figure 3B,D*).

During a glucose-to-galactose shift, *GAL* gene induction dynamics also coincide with the resumption of rapid growth (*Figure 3C*). Within all observed microcolonies, *GAL* gene induction coincides with rapid and exponential growth, while there is little to no growth in microcolonies not inducing the *GAL* genes. In contrast to glucose-maltose shifts, we do not observe a period of slow growth before *GAL* gene induction (*Figure 3C* and *Figure 3—figure supplement 3*).

We conclude that *MAL* or *GAL* gene induction and the resumption of rapid exponential growth coincide, which is consistent with the idea that *MAL* or *GAL* gene induction is crucial for cells to resume rapid growth. However, although *MAL* or *GAL* gene induction coincides with resumption of rapid exponential growth, this does not necessarily imply that *MAL* or *GAL* gene induction is the main determinant of lag phase duration and HDB. In fact, the fact that lag duration does not correlate with Gal and Mal protein levels, and the slow growth before *MAL* gene expression suggests that other mechanisms may be the main rate-limiting step for cells escaping the lag phase (*Figure 3*).

## Effect of constitutive *MAL* gene expression on HDB

So far, our results suggest that *MAL* gene induction is the final, but not the only, event that controls full escape from the lag phase and that the rate of *MAL* induction does not depend on Mal protein inheritance. To investigate this further, we measured HDB in strains with single or dual overexpression of the genes encoding the maltose-cleaving enzyme Mal12p and the transporter Mal11p (*Figure 3—figure supplements 4* and *5*). In these experiments, HDB was measured after pre-growth in either maltose or galactose. After maltose pre-growth, we observed the combined effects of constitutive expression and protein inheritance of the Mal proteins that are expressed under their native promoters. After galactose pre-growth, we only observe the effect of constitutive expression.

If induction of the Mal proteins is the crucial step allowing escape from a glucose-to-maltose lag phase, constitutive overexpression of the maltose importer Mal11p and the maltase Mal12p in glucose should reduce the lag to a minimum and abolish the HDB in glucose-to-maltose transitions. As expected, we find that the dual *MAL11* and *MAL12* overexpression strain indeed shows a strongly reduced lag phase (*Figure 3—figure supplements 4* and *5*). In contrast, this is not the case when only one of the *MAL* genes is overexpressed (*Figure 3—figure supplements 4* and *5*).

It is important to note that our overexpression strains also still contain a copy of the *MAL* genes under control of their native promoter. Thus, to only observe the effect of overexpression, and remove the potential effect of protein inheritance of the other *MAL* gene expressed under its native promoter, we should consider the results after pre-growth on galactose (*Figure 3—figure supplement 5*). After galactose pre-growth, overexpression of *MAL12* strongly increases lag time, while overexpressing *MAL11* slightly reduces the lag time. This result is consistent with their roles in positive and negative feedback loops in the *MAL* regulatory network: the transporter Mal11p would increase the intracellular concentration of maltose, which acts as an inducer of the network, whereas overexpression of the maltase Mal12p lowers intracellular maltose concentrations by enzymatic breakdown.

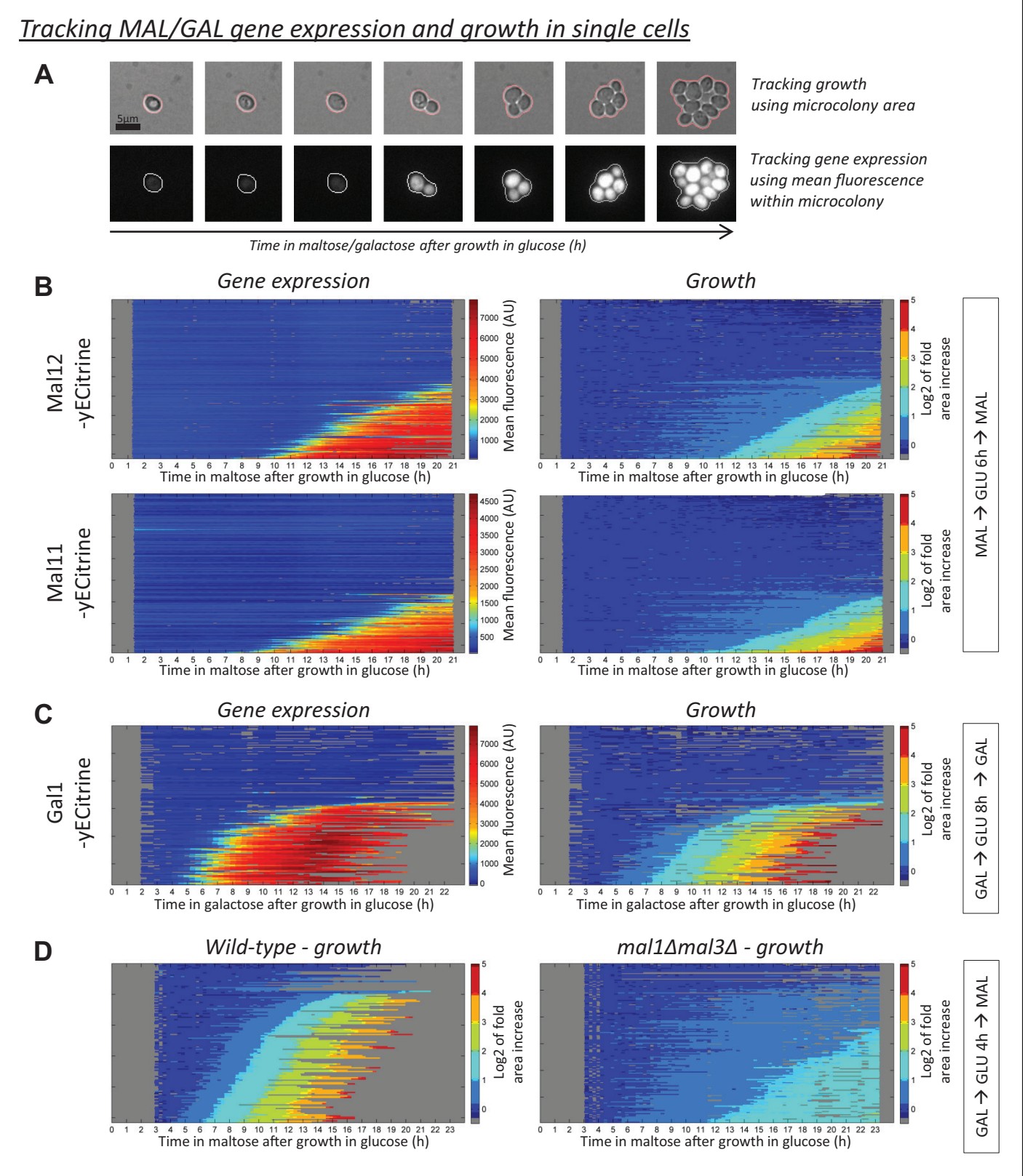

Figure 3. *MAL(GAL)* gene induction correlates with escape from the glucose-to-maltose(galactose) lag phase. (**A**) Example of tracking of the area and mean fluorescence of a microcolony. All tracked microcolonies initially started out as a single (non-)budding cell. (**B**) Kymographs showing the evolution of (left) mean Mal12-, Mal11-yECitrine fluorescence and (right) the area of the tracked microcolonies. Cells were pre-grown in maltose, switched to
*Figure 3 continued on next page*

*Figure 3 continued*

glucose for 6 hr, and back to maltose. Each horizontal line represents one microcolony, with the x-axis representing time after the shift to maltose. The microcolony tracks were sorted by their area profiles. (C) Kymographs showing the evolution of (left) mean Gal1-yECitrine fluorescence and (right) the area of the tracked microcolonies. Cells were pre-grown in galactose, switched to glucose for 8 hr, and back to galactose. (D) Microcolony growth for the wild-type and a strain which has all *MAL* genes deleted. Cells were pre-grown in galactose, switched to glucose for 4 hr, and back to maltose.

DOI: https://doi.org/10.7554/eLife.39234.014

The following figure supplements are available for figure 3:

**Figure supplement 1.** Correlation between the time of Mal12 induction and the fold area increase that occurred until this event.
DOI: https://doi.org/10.7554/eLife.39234.015

**Figure supplement 2.** Correlation between the time of Mal11 induction and the fold area increase that occurred until this event.
DOI: https://doi.org/10.7554/eLife.39234.016

**Figure supplement 3.** Correlation between the time of Gal1 induction and the fold area increase that occurred until this event.
DOI: https://doi.org/10.7554/eLife.39234.017

**Figure supplement 4.** Effect of constitutive expression of the *MAL* genes on HDB with maltose pre-growth.
DOI: https://doi.org/10.7554/eLife.39234.018

**Figure supplement 5.** Effect of constitutive expression of the *MAL* genes on HDB with galactose pre-growth.
DOI: https://doi.org/10.7554/eLife.39234.019

**Figure supplement 6.** A mating assay shows that, after short periods of glucose growth, cells contain factor(s) which promote rapid growth re-initiation.
DOI: https://doi.org/10.7554/eLife.39234.020

**Figure supplement 7.** A mating assay indicates a cytoplasmic step which controls the rate of *MAL* gene induction.
DOI: https://doi.org/10.7554/eLife.39234.021

Constitutively high levels of both Mal12p and Mal11p are sufficient to allow rapid resumption of growth after the shift to maltose (*Figure 3—figure supplements 4* and *5*). However, we also found that there is no correlation between lingering Mal11p and Mal12p proteins prior to a glucose-to-maltose shift in a wild-type strain (*Figure 2*). Therefore, we hypothesized that in the wild-type strain, at least one of these proteins is inherited in a non-functional form during glucose growth. Previous reports indeed suggest that Mal11 is rapidly internalized and inactivated after a shift from maltose to glucose (*Görts, 1969*; *Horak and Wolf, 1997*; *Jiang et al., 1997*; *Lucero et al., 2000*; *Novak et al., 2004*; *Riballo et al., 1995*).

To investigate this further, we measured HDB of the single *MAL11* and *MAL12* overexpression strains after pre-growth in maltose (*Figure 3—figure supplement 4*). We then see the effect of *MAL* overexpression of one gene and protein inheritance of the other gene expressed under its native promoter. If the natively expressed Mal protein is inherited in a functional form, it should be able to complement the other constitutively expressed Mal protein leading to a short lag time. We would actually expect the lag time to be particularly short after short glucose periods, when the inherited protein levels are still high. This is in fact the case for the *MAL11* overexpression strain, where inheritance of native Mal12p elicits minimally short lag times after short glucose periods (*Figure 3—figure supplement 4*). However, for the *MAL12* overexpression strain, there seems to be little to no effect of native Mal11 protein inheritance, even in a maltose-glucose-maltose transition where the glucose growth is limited to only 2 hr (*Figure 3—figure supplement 4*), suggesting that even if Mal11 proteins are present during glucose growth, they are not able to import maltose, most likely because they have been internalized.

In summary, Mal11p and Mal12p levels have the potential to strongly influence the lag time only if both are present at sufficiently high levels and in an active form. This requirement is not fulfilled in wild-type cells because only the maltase Mal12p is present at high enough concentrations and/or in an active form after glucose growth.

## HDB depends on a growth-promoting factor

So far, our results show that the presence of high levels of both the Mal11 maltose transporter and Mal12 maltase are sufficient for cells to escape the lag phase. However, the results also show that Mal protein inheritance is not the cause of HDB during glucose-to-maltose shifts. A next set of experiments is therefore aimed at determining what is the actual cause of the HDB. HDB must be caused by at least one factor that gradually changes during glucose growth. This factor could be a

promoting factor and enhance *MAL* gene induction and growth resumption, in which case its concentration would gradually diminish during growth in glucose. Alternatively, it could be a repressor and its concentration would increase during growth in glucose, thereby gradually increasing its repression of *MAL* gene induction and growth resumption.

To distinguish between these two scenarios, we measured the lag phase of diploid cells generated by mating haploid cells pre-grown in separate cultures (*Figure 3—figure supplement 6*). These two haploid cultures are mixed and grown in glucose for 4 hr, during which a fraction of the cells mate and form diploid cells. This mixed culture is transferred to maltose and the lag phase of the individual cells is scored. When the two parents are grown on the same pre-growth medium (either both in glucose or both in maltose), the resulting diploids and the haploid parental populations demonstrate the same lag behavior (*Figure 3—figure supplement 6*, top panels). Interestingly, when one parent has been pre-grown on maltose and the other parent is pre-grown on glucose, the resulting diploid demonstrates a lag similar to the parent pre-grown on maltose, and thus has a short lag phase. (*Figure 3—figure supplement 6*, bottom panels). This suggests that the diploid has received a high concentration of a growth-promoting factor from the parent that has been grown on maltose. In other words, this result suggests that the factor determining HDB in glucose-maltose shifts promotes growth resumption and is gradually reduced during glucose growth.

## HDB in *MAL* gene induction is independent from nuclear-retained factors

Now that we have evidence to support the presence of growth-promoting factor(s) and that *MAL* gene induction is crucial for complete growth resumption, we investigated whether the events leading up to *MAL* gene induction are restricted to the nucleus or if they involve at least one cytoplasmic event. We performed a similar mating assay as the experiment described above with a slight alteration: one of the parents contains a *kar1-1* allele, which allows for cytoplasmic fusion but prevents nuclear fusion during the mating process. Mating the *kar1-1* parent with a *KAR1*-containing parent generates a heterokaryon instead of a regular diploid (*Zacharioudakis et al., 2007*). Both parents also contain a *MAL12* allele fused to a differently colored fluorescent protein so we could measure the start of *MAL* gene induction in the two different parental nuclei. We found that the rate of *MAL* gene induction is not biased towards one of the two nuclei in the heterokaryon cells (*Figure 3—figure supplement 7*), indicating that the events leading up to *MAL* gene induction are not restricted to the nucleus, but must involve at least one cytoplasmic step. Importantly, this argues against the hypothesis that HDB is exclusively due to a transcriptionally poised chromatin structure at the *MAL* genes.

## Genome-wide screen identifies distinct sets of genes that influence HDB

Our results thus far indicate that while *MAL* gene induction is needed for complete adaptation to maltose, HDB after a glucose-to-maltose shift cannot be ascribed to factors directly related to the *MAL* regulatory network. To unravel the mechanism underlying the HDB, we performed a genome-wide Bar-Seq screen to identify gene deletion mutants that demonstrate altered HDB (*Figure 4*). Bar-Seq experiments provide the ability to track growth characteristics of thousands of individual gene deletion mutants through the sequencing of unique barcodes present in each individual mutant (*Robinson et al., 2014*; *Smith et al., 2009*). The experimental set-up and growth of the pooled populations is shown in *Figure 4—figure supplement 1* and described in the Materials and Methods section.

We characterized maltose-glucose-maltose HDB in 3548 gene deletion mutants. To represent the dynamics of HDB of mutants deleted for individual genes, we plotted the lag behavior after different times in glucose (*Figure 4A* and *Figure 4—figure supplement 3*). These graphs represent the HDB of each of the 3548 mutants. The procedure for calculating these lag times from read counts for each biological replicate is detailed as *supplementary file 1*. The graphs consist of an upper and a lower panel. When the lag time after a certain time in glucose could be quantified for a mutant, this lag time is plotted on the lower panel. If, however, we could not detect a clear escape within the timeframe of the experiment, we use the upper panel to plot the (small) relative change in cell density of the mutant during the lag phase. Using this approach, we identified four types of mutants

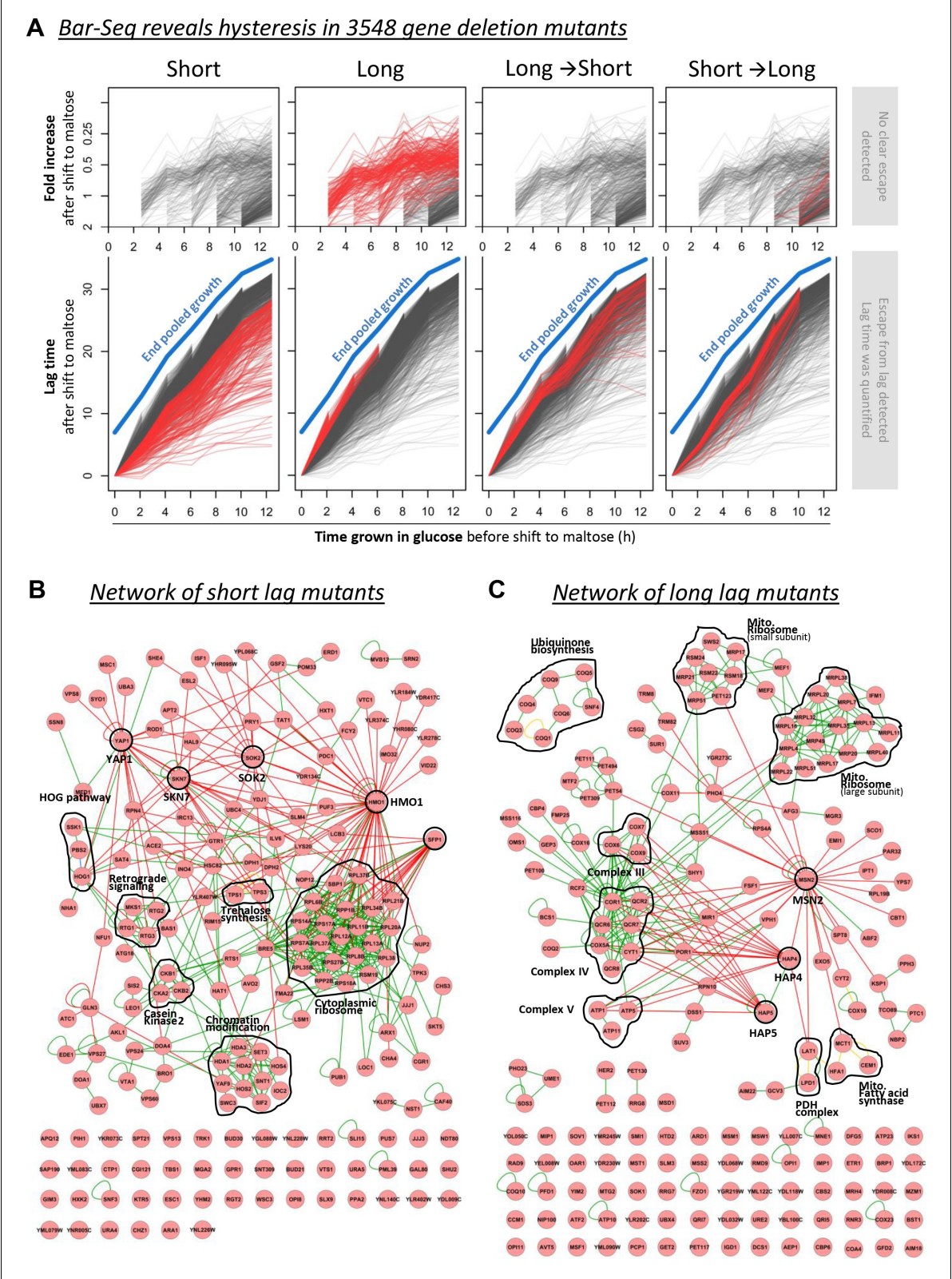

**Figure 4.** Barcode sequencing HDB dynamics in 3548 mutants. (**A**) The black lines show the lag durations of all mutants for different times of pre-growth in glucose. The red lines indicate four types of mutants with (1) generally shorter lags, (2) generally longer lags, (3) relatively long lags after short glucose periods, and *vice versa*, (4) relatively short lags after short glucose periods, but long after long glucose periods. The quantification and visualization of these results are explained in more detail in ***Figure 4—figure supplement 2***. (**B**) A physical interaction network for the genes identified

*Figure 4 continued on next page*

*Figure 4 continued*

as generally leading to shorter lags upon deletion. It includes DNA-protein (red lines), protein-protein (green lines) and phosphorylation (yellow lines) interactions. (C) Same as in (B), but for the genes identified as generally leading to longer lags upon deletion.

DOI: https://doi.org/10.7554/eLife.39234.022

The following figure supplements are available for figure 4:

**Figure supplement 1.** Experimental set-up of the Bar-Seq experiment.

DOI: https://doi.org/10.7554/eLife.39234.023

**Figure supplement 2.** Example of the growth rate and lag time analysis for one mutant.

DOI: https://doi.org/10.7554/eLife.39234.024

**Figure supplement 3.** Example visualization of HDB for one mutant.

DOI: https://doi.org/10.7554/eLife.39234.025

**Figure supplement 4.** A physical interaction network for the genes identified as having relatively long lags after short glucose periods, and *vice versa*.

DOI: https://doi.org/10.7554/eLife.39234.026

**Figure supplement 5.** A physical interaction network for the genes identified having relatively short lags after short glucose periods, but long after long glucose periods.

DOI: https://doi.org/10.7554/eLife.39234.027

**Figure supplement 6.** Validation of the BAR-Seq experiment.

DOI: https://doi.org/10.7554/eLife.39234.028

based on their lag behavior (*Figure 4A*). Two large groups contain mutants that consistently show either a shorter or a longer lag phase compared to wild-type cells. The interaction networks for the genes of these two groups are shown in *Figure 4B and C*, respectively. A third category of mutants demonstrates a long lag time after short glucose periods, but average-to-short lags after long glucose periods. A fourth set of mutants show the opposite behavior compared to the third group, with short lags after short glucose periods and long lags after long glucose periods. The interaction networks for the genes of the latter two groups are shown in *Figure 4—figure supplements 4* and *5*.

In order to validate the results from the Bar-Seq, we investigated the effect of three genes, *ATP5*, *COQ5* and *QCR7*, that were identified in the Bar-Seq experiment as determinants of lag duration and HDB (*Figure 4—figure supplement 6*). We therefore created three mutant cells lines, each lacking one of the three candidate genes. Similar to the setup of the Bar-Seq experiment, cultures were pre-grown in maltose, then shifted to glucose for different durations, and washed back to maltose and the population lag phase was measured. Consistent with the Bar-Seq, mutants in which one of these three genes was deleted have a longer lag phase compared to the wild-type.

Using gene ontology (GO) analysis, we identified functional enrichment in the four types of mutants. Deletion mutants with shorter lag phases are enriched for transcriptional regulation, the cytoplasmic ribosome, and histone deacetylation, suggesting that these mutants are defective in gene regulation associated with glucose repression (*Figure 4B*). More surprisingly, deletion mutants with longer lag phases are highly enriched for genes involved in respiration and mitochondrial function (*Figure 4C*). The long-then-short mutants contain members of the chromatin modifiers SAS- and SET1/COMPASS-complexes. The short-then-long mutants are enriched for de-novo purine biosynthesis and ESCRT-complexes that are involved in endocytosis (*Figure 4—figure supplements 4* and *5*).

## Transcriptome analysis reveals regulation of genes involved in respiratory metabolism

To further dissect the molecular mechanisms underlying HDB, we used RNA-Seq to measure the transcriptional changes that occur during glucose growth and during the lag phase after the shift to maltose. We reasoned that genes that are downregulated during glucose growth and induced early during the lag phase after a shift to maltose might be important in determining lag phase length. The experimental set-up and the sampling scheme are shown in *Figure 5A* and are described in the Materials and Methods section. Expression values for biological replicates at each sampling point are provided as a *supplementary files 2* and *3*.

After sequencing, samples were compared with the maltose pre-growth condition to detect genes that were significantly up- or down-regulated relative to this condition. For each sample, GO enrichment was performed to link these genes to biological processes. Finally, we selected a set of

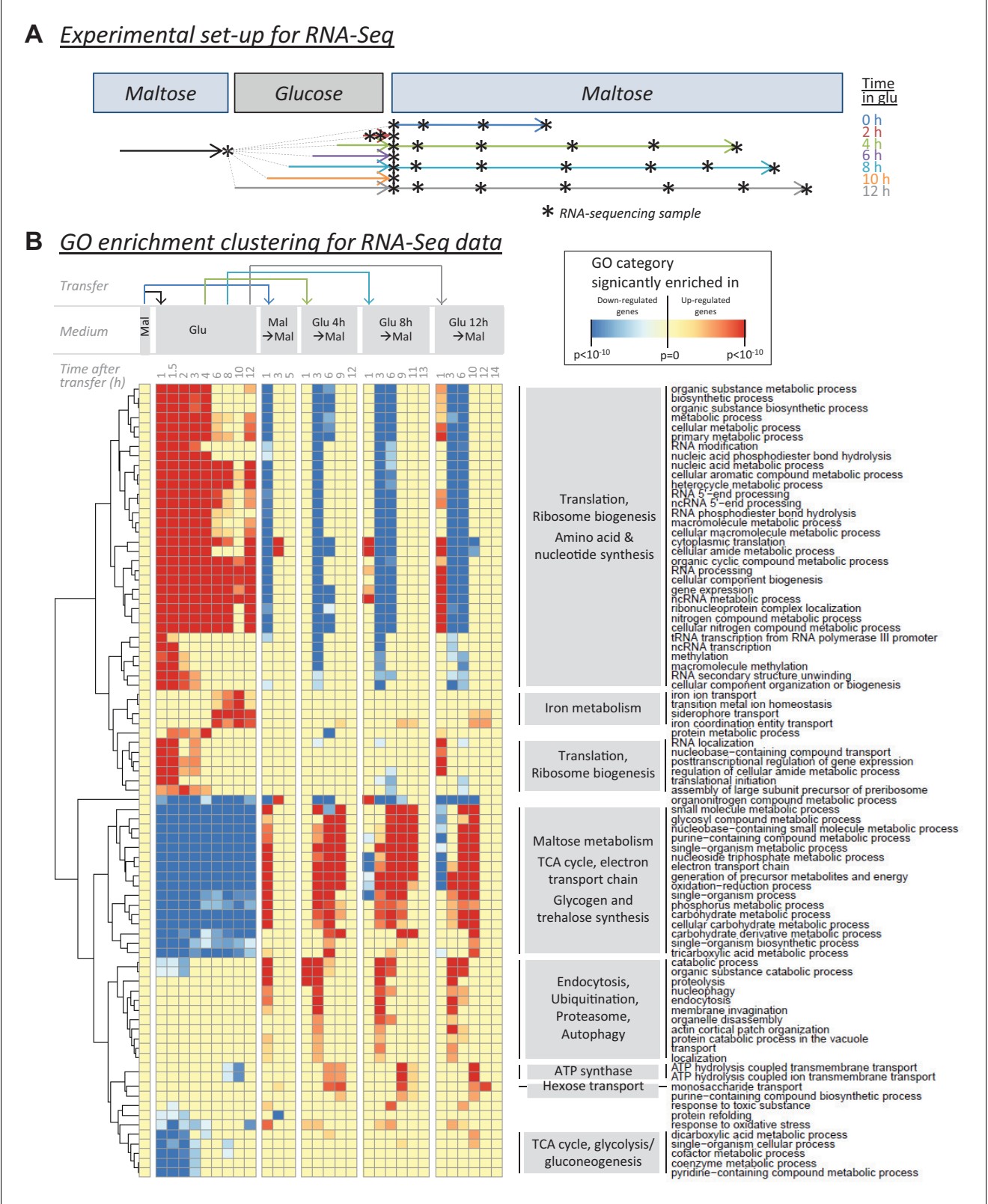

**Figure 5.** RNA-sequencing reveals cellular functions that are regulated during glucose and the lag phase. (**A**) Experimental set-up of the RNA-Seq experiment. The wild-type strain adapted to maltose growth was transferred to glucose for different times (0, 2, 4, 6, 8, 10, 12 hr) before switching it back to maltose. Multiple samples (indicated by stars) were taken during the different growth phases. The final sample was taken when the cultures increased 4-fold after the shift to maltose. (**B**) Heatmap representing a selection of non-redundant GO categories that were significantly enriched in the

*Figure 5 continued on next page*

*Figure 5 continued*

set of up- or downregulated genes of at least three samples throughout the experiment (Materials and Methods). The heatmap displays the GO category on the horizontal axis, the sample on the vertical axis, while the color scale indicates enrichment of the GO category. The clustering by rows reveals groups of related GO categories which are indicated on the right in grey boxes.

DOI: https://doi.org/10.7554/eLife.39234.029

The following figure supplements are available for figure 5:

**Figure supplement 1.** Detection of genes that show large transcriptional changes between 1 hr and 12 hr after the shift to glucose.

DOI: https://doi.org/10.7554/eLife.39234.030

**Figure supplement 2.** Log2 expression changes during glucose growth for the genes that show large transcriptional changes between 1 hr and 12 hr after the shift to glucose Log2 expression of the genes highlighted in *Figure 5—figure supplement 1* is shown between 1 hr and 12 hr after the shift to glucose.

DOI: https://doi.org/10.7554/eLife.39234.031

**Figure supplement 3.** Correlation between the lag behavior upon gene deletion (Bar-Seq), and the expression change of the corresponding genes during glucose growth (RNA-Seq).

DOI: https://doi.org/10.7554/eLife.39234.032

**Figure supplement 4.** Correlation between protein and mRNA level changes during glucose growth after pre-growth on maltose.

DOI: https://doi.org/10.7554/eLife.39234.033

non-redundant GO categories that were identified in at least three samples throughout the experiment (*Figure 5B*).

Perhaps not surprisingly, genes associated with growth, such as genes encoding elements of the translation machinery are up-regulated during glucose growth and repressed during the lag phase following a shift to maltose. Conversely, the *MAL* genes are repressed during glucose growth and induced during the lag phase following the shift to maltose. Interestingly, genes involved in respiration-related pathways, such as the TCA cycle and the electron transport chain, mirror *MAL* gene regulation: they are repressed during glucose growth and induced after the shift to maltose. Moreover, the induction of these respiration-linked genes actually precedes the induction of the *MAL* genes. Apart from these two large groups, we also identified specific processes that are uniquely up-regulated in glucose or after the shift to maltose. There is a strong induction of genes involved in iron metabolism at the end of glucose growth (*Figure 5B* and *Figure 5—figure supplements 1* and *2*) and processes such as endocytosis, autophagy and proteolysis are up-regulated after the shift to maltose (*Figure 5B*).

We were particularly interested in genes that demonstrated different changes in expression after short and long pre-growth in glucose as these are likely candidates to be involved in HDB. We compared the mean change in expression and the rate (slope) of change in expression and were able to identify genes for which the expression level changed significantly and gradually with increasing times of glucose growth (*Figure 5—figure supplements 1* and *2*). Interestingly, among the genes with a gradual repression during glucose (*Figure 5—figure supplement 1*, bottom left quadrant), we find *HAP4*, a central regulator of respiration (*Lascaris et al., 2003*; *Lascaris et al., 2004*), again hinting at a central role for respiration during the lag phase.

We compared the RNA-Seq results with the Bar-Seq results and asked whether genes that have a large effect on the lag time upon deletion show specific expression changes in glucose. In general, we find that genes that lead to a long lag when deleted tend to be repressed during glucose growth (repressed on average by 35%), whereas genes that lead to a shorter lag when deleted are on average slightly induced during glucose growth (induced on average by 10%) (*Figure 5—figure supplement 3*). Since we know that the factor determining HDB promotes escape from the lag and is reduced during glucose growth, this first category of repressed genes, which mainly contains genes linked to respiration, is likely linked to HDB.

In order to test to what extent the observed trends in mRNA levels hold true for protein levels, we compared the change in protein levels and the change in mRNA levels during glucose, growth after pre-growth in maltose, for a selected set of key genes and proteins (*Figure 5—figure supplement 4*). The protein levels were measured using flow cytometry on strains with genomically integrated fluorescent protein fusions. For this purpose, we chose six respiratory genes (*CIT1*, *MDH1*, *KGD2*, *NDI1*, *SDH2* and *COX6*) alongside the *MAL11* and *MAL12* genes. The results show that in general, there is a good correlation between changes in mRNA levels and protein levels.

## Levels of respiratory proteins correlate with population lag time

The combined results from our RNA-Seq and Bar-Seq screens suggest that genes involved in respiratory metabolism play a central role in the glucose-to-maltose transition. This is particularly interesting for *S. cerevisiae* as it is a Crabtree-positive yeast, meaning that glucose represses respiration in favor of fermentation even in the presence of oxygen. This behavior resembles the Warburg effect observed in mammalian tumors (*Vander Heiden et al., 2009*; *Warburg, 1956*).

We hypothesized that when cells are shifted from maltose to glucose, the cells reduce their respiratory metabolism, but do so gradually (*Slavov et al., 2014*). When cells are shifted from glucose to maltose, they need to re-induce respiration to continue growth. This implies that cells demonstrating higher respiration activity while growing on glucose would have shorter lag times upon a shift to maltose or galactose. We would therefore expect a negative correlation between the level of respiratory proteins in glucose and the lag time.

To test this hypothesis, we used flow cytometry to track the presence of a set of respiratory proteins during glucose growth after pre-growth in either maltose or galactose (*Figure 6A*). Moreover, we measured the level of these proteins during lag phase using fluorescence microscopy (*Figure 6—figure supplement 1*). We chose key proteins of the TCA cycle (Cit1p, Mdh1p and Kgd2p) and the electron transport chain (Ndi1p, Sdh2p and Cox6p). The expression levels of these proteins have been shown to correlate with the degree of respiration (*Fendt and Sauer, 2010*). Their levels correlate well with the population lag times, for pre-growth in maltose and also for pre-growth in galactose, suggesting that the link between respiration and lag duration is a general phenomenon ($R^2$ between 0.76 and 0.97) (*Figure 6A*). Moreover, in contrast to the correlation between population lag times and Mal protein levels (*Figure 2E*), the correlation between respiratory activity and lag times upon a switch to maltose holds after pre-growth in maltose, as well as after a pre-growth in galactose (*Figure 6A*. Also, consistent with the hypothesis that the ability to induce respiration decreases with residence time in glucose, we observe that the transcript levels of COX6 and CIT1 respiratory genes during lag phase is induced to its peak earlier in cells with shorter pre-growth in glucose (*Figure 6—figure supplement 2*).

If respiration is an important determinant of HDB, we would expect a clear link between gene expression dynamics and the escape from the lag phase after a shift from glucose to maltose. Using our microcolony tracking set-up (*Figure 3A*), we found that only microcolonies that induce these respiratory genes show substantial growth (*Figure 6—figure supplement 1*). Interestingly, the induction of these genes occurs around the same time as the start of the slow growth period observed within many microcolonies and precedes the induction of the *MAL* genes by several hours (*Figure 3B* and *Figure 6—figure supplement 1*). To better demonstrate this point, the induction of these respiratory proteins as well as MAL11 maltose transporter and MAL12 maltase during the lag phase was compared to the growth dynamics of the cells. The time-resolved cross-correlation between the growth rate and the protein expression was calculated for a range of time offsets (see (*Kiviet et al., 2014*) and *Figure 6D*). The cross-correlation for Mal11p and Mal12p peaks right after the cells resume growth. The correlation for Cit1p and Cox6p, however, peaks around 4 hr earlier and is stronger, indicating that expression of these respiratory proteins precedes escape from the lag phase, whereas, on average, expression of the maltose proteins only occurs around the time of escape.

## Respiratory activity determines HDB

To investigate if respiratory activity directly affects the lag phase and HDB, we altered the cells' respiratory capacity and measured HDB after a glucose-to-maltose shift. We used a chemical inhibitor of respiration (antimycin A) to decrease respiration as the cells transition from glucose to maltose. Furthermore, we examined a respiratory deficient mutant (*cox6Δ*). For both methods, altering the respiratory capacity led to substantially longer lag phases and a stronger increase in lag with extended pre-growth in glucose (*Figure 6B*). Conversely, to increase respiratory activity during glucose growth and the lag phase, we overexpressed the transcription factor *HAP4* (*Lascaris et al., 2003*; *Lascaris et al., 2004*). On average, the *HAP4* overexpression strain showed a two-fold decrease in lag duration and a much smaller increase in lag with extended pre-growth on glucose. This effect occurs both after pre-growth in maltose and pre-growth in galactose (*Figure 6C*). Measurement of dissolved oxygen during lag phase and glucose growth indeed confirms that *HAP4*

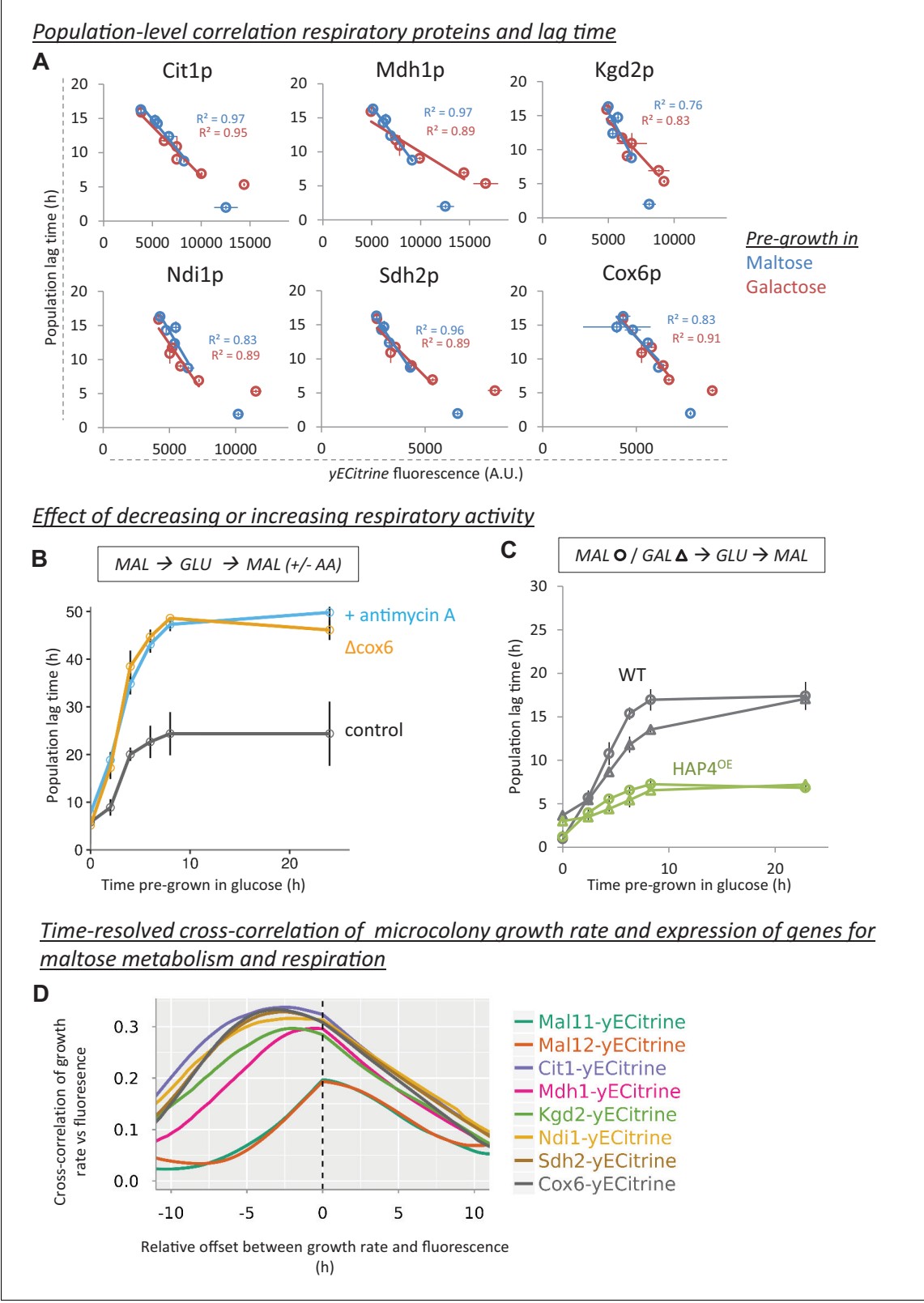

**Figure 6.** Decrease of respiratory protein levels during glucose growth correlate with HDB. **(A)** Population-level correlation between respiratory protein levels of Cit1-, Mdh1-, Kgd2-, Ndi1-, Sdh2- and Cox6-yECitrine and the lag time. Lag times and respiratory protein levels are correlated after pre-growth in maltose or galactose. The cultures that did not experience glucose (steady-state maltose or galactose) were not included in the correlations. The error bars for flow cytometry represent the range for two replicates. **(B)** The effect of decreasing respiratory activity on HDB. Two methods were

*Figure 6 continued on next page*

*Figure 6 continued*

used to lower respiratory activity: chemical inhibition by antimycin A (μg/ml) added after shift to maltose and a respiratory mutant strain (cox6Δ). The error bars represent SEM for three replicates. (**C**) The effect of increasing respiratory activity on HDB. An overexpression strain of *HAP4* showing increased respiratory capacity was compared to the wild-type. Both strains were pre-grown in either maltose (circles) or galactose (triangles) before the transfer to glucose, and finally maltose. The error bars represent the range for two replicates. (**D**) Induction of respiratory proteins, but not maltose genes, is an early predictor of escape from the lag phase. Expression of Mal11-, Mal12-, Cit1-, Mdh1-, Kgd2-, Ndi1-, Sdh2- and Cox6-yECitrine is quantified in single cells which will grow into microcolonies through the lag phase. The graphs show the time-resolved cross-correlation between microcolony growth rate and protein expression. For Cox6-yECtirine and Cit1-yECtirine, the cross-correlation shows a peak around 5 hr, meaning that in average, expression of these proteins anticipates growth resumption 5 hr later. However, for maltose genes the peak is located slightly around 0 hr, which implies that induction of these proteins coincides with growth resumption. Number of the cells is detailed in the methods section.
DOI: https://doi.org/10.7554/eLife.39234.034

The following figure supplements are available for figure 6:

**Figure supplement 1.** Tracking respiratory gene induction during the lag phase.
DOI: https://doi.org/10.7554/eLife.39234.035
**Figure supplement 2.** Induction of respiratory genes for different glucose pre-growth times.
DOI: https://doi.org/10.7554/eLife.39234.036
**Figure supplement 3.** Oxygen consumption during glucose growth and lag phase.
DOI: https://doi.org/10.7554/eLife.39234.037

overexpression and respiratory deficient mutant (*cox6Δ*) strains consume oxygen at higher and lower rates respectively, compared to the wild-type (*Figure 6—figure supplement 3*).

At first sight, the effect of *HAP4* overexpression on lag duration and HDB may appear similar to dual *MAL11* and *MAL12* overexpression, as both lead to reduced lag duration and a severe reduction in HDB. However, while activation of the M AL genes is undoubtedly the final crucial step that allows cells to escape the lag phase, our results suggest that in contrast to what is often suggested, the induction of *MAL* genes is not the rate-limiting step and that inheritance of Mal proteins or *MAL* gene expression status does not explain HDB. Activation of respiration actually precedes *MAL* gene induction, and this activation of respiratory metabolism is the main rate-determining step. Together, these results show that the level of respiratory activity has a strong, direct influence on the duration of the lag time and the associated HDB.

## Discussion

In this study, we used single-cell analyses and genome-wide screens to dissect the molecular mechanism underlying the HDB that affects lag phase length after a shift in carbon source. Our experiments show that cells pre-grown on maltose before being shifted to glucose and then back again to maltose show a lag phase that depends on the time the cells spent in glucose.

HDB is often implicitly suggested to be caused by changes in the dynamics of the de-repression of specific genes. This hypothesis is often based on a correlation between the induction of certain genes and the resumption of growth, for example in shifts from glucose to galactose. In this case, the HDB has been proposed to be due to Gal protein inheritance. In this work, we provide several lines of evidence indicating that HDB in maltose-glucose-maltose shifts depends neither on the first encounter with maltose nor on inheritance of Mal proteins. Instead, the HDB is linked to a previous exposure to any alternative carbon source that requires some degree of respiration, and consequently the gradual decline of respiratory activity during growth on glucose.

HDB does not depend on Mal protein inheritance. Firstly, intracellular Mal protein levels do not correlate with the population lag times across different pre-growth conditions. Specifically, we showed that the HDB that occurs when cells are shifted from glucose-to-maltose or glucose-to-galactose does not depend on a previous exposure to maltose or galactose, but rather on the time that the population was grown in glucose prior to the shift to the alternative carbon source (*Figure 2A–F*). Secondly, the variability in Mal protein levels within cells from one population does not explain the heterogeneity in lag duration between single cells (*Figure 2G–I*). Thirdly, only high levels of both functional maltase (Mal12p) and maltose transporter (Mal11p) can affect HDB (*Figure 3—figure supplements 4* and *5*). The activity or concentration of the transporter appears to be too low to influence the lag time, even after only 2 hr of growth in glucose. This observation agrees

with previous reports that the maltose transporter is rapidly internalized by endocytosis and degraded in the vacuole in the presence of glucose (*Görts, 1969*; *Horak and Wolf, 1997*; *Jiang et al., 1997*; *Lucero et al., 2000*; *Novak et al., 2004*; *Riballo et al., 1995*). Fourthly, while expression of *MAL* genes is required for cells to resume fast exponential growth (*Figure 3B,D* and *Figure 3—figure supplements 4* and *5*), *MAL* gene induction does not seem to be the primary bottleneck determining lag phase duration and thus HDB. Tracking microcolony growth and gene expression after the glucose-maltose shift showed that cells that eventually induce the *MAL* genes, often show a transient, slow growth phase before they induce these genes (*Figure 3B* and *Figure 3—figure supplements 1* and *2*). Importantly, this slow growth is still observed in a mutant in which both the maltose transporter and the maltase genes were deleted (*Figure 3D*). Together, these results suggest *MAL*-independent processes that precede and possibly stimulate *MAL* gene induction.

To identify processes that could precede *MAL* gene activation and thereby determine HDB, we performed two genome-wide screens. This allowed us, for the first time, to obtain unparalleled insight into the molecular mechanisms underlying HDB. In both Bar-Seq and RNA-Seq experiments, genes involved in mitochondrial function and respiration demonstrated strong effects. Deletion mutants of genes involved in mitochondrial function and respiration showed longer lag times (*Figure 4C*). Moreover, the same genes were repressed during glucose growth (*Figure 5—figure supplements 2* and *3*; and *Figure 6A*). We confirmed that respiratory activity directly influences the lag phase by testing a respiratory deficient mutant as well as chemically inhibiting respiration, (*Figure 6B* and *Figure 6—figure supplement 3*) and on the other hand by increasing respiratory activity using *HAP4* overexpression (*Figure 6C* and *Figure 6—figure supplement 3*; *Lascaris et al., 2004*). These experiments confirmed that changes in the respiratory activity cause changes in HDB. Specifically, reduced respiratory activity results in longer lag phases and over-activation of respiration yields shorter lag phases. Moreover, the slow growth observed in cells that just escaped the lag phase only occurs in microcolonies that induce respiratory proteins, with slow growth starting around the same time as respiratory gene induction, hours before *MAL* gene induction is observed (*Figure 3B* and *Figure 6—figure supplement 1*)

Our hypothesis that HDB is due to a gradual decrease in respiratory activity during glucose growth is consistent with a previous report demonstrating a gradual decrease of respiration over multiple generations in glucose (*Slavov et al., 2014*) as well as with a recent study showing the importance of respiration for a switch from glucose to other carbon sources (*Perez-Samper et al., 2018*). Our results from the mating and heterokaryon experiments are also consistent with this hypothesis. The mating experiment indicated that HDB is caused by a growth-promoting factor that is diluted during glucose growth. A caveat in the mating experiment is that after mating, the proteins uniquely present in each of the two cells are diluted two-fold which might render a repressor non-functional. However, this does not seem likely given the graded HDB for different glucose pregrowth times. The heterokaryon experiment showed that the rate of *MAL* gene induction is determined by at least a cytoplasmic process (*Figure 3—figure supplements 6* and *7*). The gradual decrease in respiratory activity could be either directly due to protein inheritance of respiratory proteins or complexes during glucose growth, and/or indirectly by the gradual change in an upstream factor determining respiratory activity independently of the respiratory protein levels. Our population-level lag times correlated well with respiratory protein levels, supporting the former hypothesis. Moreover, this could explain the observation that pre-growth in galactose leads to shorter lag times compared to pre-growth in maltose as the levels of respiratory proteins are higher in galactose than in maltose. (*Figure 2A–B* and *Figure 6A*).

Why would respiration play such an important role during the shift from glucose to alternative sugars? We hypothesize that upon the sudden loss of glucose, cells enter an energy-deficient state because they are not yet able to metabolize maltose. This energy-deficient state would be exacerbated by a decreased respiratory activity in response to glucose: respiration is a more energy-efficient pathway per carbon molecule compared to fermentation. Finally, this energy-deficient state likely prevents *MAL* gene induction and subsequent escape from the lag. In cells that successfully induce the *MAL* genes, it is unlikely that *MAL* gene expression is prevented by the complete depletion of factors that are required for protein expression (e.g. ATP). Indeed, in these cells, protein synthesis of other proteins occurs well before synthesis of Mal proteins. However, it is possible that for the majority of cells that never resume growth, protein expression is inhibited altogether by the

depletion of necessary factors. Indeed, these cells show no sign of any protein synthesis (*Figure 3B* and *Figure 6—figure supplement 1*). An intriguing possibility is that in the cells that will eventually induce the *MAL* genes, *MAL* gene induction might be prevented until the cells have at least partly recovered from the dip in metabolic flux and ATP production before the energetically costly proton gradient over the cell membrane is used to support maltose import via the H⁺-maltose symporter (*Serrano, 1977*). The hypothesis that respiration is required for bridging the energy-deficient state during the lag phase is in agreement with the observation that lag phase in maltose is similar for cultures with pre-growth in maltose or galactose since respiration is active during growth on both sugars and repressed during glucose growth.

Previous investigations into HDB in carbon sources have not identified the general metabolic state as a central determinant. More commonly, such studies have noted inheritance of the corresponding metabolic proteins directly involved in either the uptake or processing of the alternative sugars (*Kundu and Peterson, 2010*; *Stockwell and Rifkin, 2017*; *Zacharioudakis et al., 2007*). Such studies, however, focused on the induction of the genes directly involved in alternative sugar metabolism without showing that induction of these genes is the rate-limiting step in adaptation, and without screening more broadly for processes that may precede and control induction of these genes. Although these studies revealed a correlation between residual Gal1p levels and the lag duration, they did not show that this correlation implied causation. It should, however, be noted that the experimental setups differ between studies, which may influence the main determinants of the duration of lag. Finally, the role of respiration in previously published omics expression studies on diauxic shifts (*Murphy et al., 2015*; *Radonjic et al., 2005*; *Zampar et al., 2013*), might have been masked by the fact that in their case the secondary carbon source (ethanol) can be solely utilized by respiration.

Our single-cell experiments, measuring lag time in maltose for cells that have experienced glucose repression for various durations, show that not only the lag time depends on glucose growth time, but also the fraction of cells resuming growth is as well dependent on the duration of previous exposure to glucose repression: upon shift to maltose after longer exposure to glucose, cells will have a longer lag phase and a smaller fraction of cells will manage to make the shift within the observed window of 24 hr. Considering the key role of respiration in the lag phase, one could speculate that after repression of respiration in glucose, some cells maintain some level of respiratory functionality while others do not. Such heterogeneity could be attributed to inheritance of a bistable positive feedback loops in regulation of respiration. Similar mechanism of inherited bistability is shown in the Lac operon of *E. coli* (*Novick and Weiner, 1957*; *Ozbudak et al., 2004*). Moreover, our results also show that although HDB is present in three different genetic backgrounds, the extent of the effect can vary greatly. We are currently dissecting the exact genetic differences that control the extent of HDB in natural yeast strains.

Our results reveal how slow transitions in central carbon metabolism cause transgenerational cellular HDB. This is not only interesting for our basic understanding of HDB, but also opens new routes in microbial biotechnology. One of the most common problems in industrial fermentations where complex substrates are used is that the microbes sometimes fail to switch between the different carbon sources and other nutrients in the medium, which leads to stuck or sluggish fermentations that cause important economic losses (*Bisson and Butzke, 2000*; *Verstrepen et al., 2004*). It will be interesting to see whether engineering strains, for example by overexpressing *HAP4*, can lead to more efficient fermentations.

In summary, our results demonstrate that HDB after a glucose-to-maltose shift does not only depend on the inheritance of Mal proteins required for the uptake and breakdown of maltose. Although activating the uptake and metabolism of alternative sugars is the last crucial step for cells to resume growth on a new carbon source, other processes precede this step and seem to be more important bottlenecks. Specifically, we find that during the switch from glucose to an alternative sugar such as maltose, cells need to activate respiratory metabolism. However, cells that have grown for extended times on glucose gradually repress respiration in favor of fermentation, and experience difficulties to re-activate respiratory metabolism, which results in much slower adaptation. We therefore conclude that HDB is not specific to a particular sugar. Instead, HDB is more general, with cells likely 'remembering' growing on any alternative carbon source that does not repress respiration as much as glucose does. It is tempting to speculate that similar slow transitions in metabolic states also govern HDB in other cell types, including some of the cases of nutritional programming

observed in mammals. In fact, the genes identified in our genome-wide screen shows overlap with a recent screen for oxidative phosphorylation in human cells (*Arroyo et al., 2016*), which could provide a good basis to explore the role of these genes in metabolic programming in higher organisms.

# Materials and methods

## Key resources table

| Reagent type (species) or resource | Designation | Source or reference | Identifiers | Additional information |
|---|---|---|---|---|
| Strain (*S. cerevisiae*) | BY4742 | PMID: 9483801 | | S288c MATalpha; his3Δ1 leu2Δ0 lys2Δ0 ura3Δ0 |
| Strain (*S. cerevisiae*) | BY4741 | PMID: 9483801 | | S288c MATa his3Δ1 leu2Δ0 met15Δ0 ura3Δ0 |
| Strain (*S. cerevisiae*) | KV1042 | PMID: 20471265 | | BY4741 MAL13::HygR-MAL63_c9 |
| Strain (*S. cerevisiae*) | KV1156 | PMID: 20471265 | | BY4742 MAL13::HygR-MAL63_c9 |
| Strain (*S. cerevisiae*) | AN62 | PMID: 24453942 | | KV1156 SAL1+ |
| Strain (S. cerevisiae) | AN63 | PMID: 24453942 | | AN62 mating type switched to MATa |
| Strain (S. cerevisiae) | KV2469 | This study | | KV1042 kar1-1 |
| Strain (S. cerevisiae) | MC4 | This study | | AN63 CIT1-yECitrine |
| Strain (S. cerevisiae) | MC19 | This study | | AN63 MDH1-yECitrine |
| Strain (S. cerevisiae) | BC55 | This study | | AN63 NDI1-yECitrine |
| Strain (S. cerevisiae) | BC56 | This study | | AN63 SDH2-yECitrine |
| Strain (S. cerevisiae) | BC60 | This study | | AN63 COX6-yECitrine |
| Strain (S. cerevisiae) | BC72 | This study | | AN63 KGD2-yECitrine |
| Strain (S. cerevisiae) | AN73 | PMID: 24453942 | | AN63 MAL12-yECitrine |
| Strain (S. cerevisiae) | AN77 | PMID: 24453942 | | AN63 MAL11-yECitrine |
| Strain (S. cerevisiae) | AN104 | PMID: 24453942 | | AN63 pGPD-MAL11 |
| Strain (S. cerevisiae) | AN105 | PMID: 24453942 | | AN63 pGPD-MAL12 |
| Strain (S. cerevisiae) | AN107 | PMID: 24453942 | | AN63 pGPD-MAL11, pGPD-MAL12 |
| Strain (S. cerevisiae) | JHH2 | This study | | KV1156 YRO2_intergentic_loc::pGPD-mCherry |
| Strain (S. cerevisiae) | JHH3 | This study | | KV1042 YRO2_intergentic_loc::pGPD-yECitrine |
| Strain (S. cerevisiae) | JHH9 | This study | | JHH2 x JHH 3 |
| Strain (S. cerevisiae) | JHH19 | This study | | KV2469 Htb2:CFP, MAL12:mCherry |
| Strain (S. cerevisiae) | JHH22 | This study | | KV1156 Htb2:CFP, MAL11:yECitrine |
| Strain (S. cerevisiae) | KV1908 | This study | | BY4741 MAL1x::KanMX, MAL3x::LEU2 |
| Strain (S. cerevisiae) | GP77 | This study | | AN63 ATP5::KanMX |
| Strain (S. cerevisiae) | GP83 | This study | | AN63 COQ5::KanMX |
| Strain (S. cerevisiae) | GP107 | This study | | AN63 QCR7::KanMX |
| Strain (S. cerevisiae) | GP89 | This study | | AN63 COX6::KanMX |

*Continued on next page*

*Continued*

| Reagent type (species) or resource | Designation | Source or reference | Identifiers | Additional information |
|---|---|---|---|---|
| Strain (S. cerevisiae) | L-1374 | PMID: 19840116 | | |
| Strain (S. cerevisiae) | BC187 | PMID: 19840116 | | |
| Recombinant DNA reagent | pKT140 | Addgene | | KanMX-yECitrine plasmid |
| Recombinant DNA reagent | pSR101 | PMID: 21468987 | | mCherry-pTEF-caURA3 |
| Recombinant DNA reagent | pYM-N15 | Euroscarf | | GPD promoter, natNT2 |
| Other | Sigma Aldrich | 1397-94-0 | 1397-94-0 | |
| Other | Maltose | Sigma Aldrich | M5885-1KG | D-(+)-Maltose monohydrate,>99% |
| Other | Galactose | Fisher Scientific | 150610010 | D(+)-Galactose,>99% |
| Other | Glycerol | Sigma Aldrich | G5516 | Glycerol,>99% |
| Other | Yeast extract | Lab M | MC001 | |
| Other | Bacteriological Peptone | Lab M | MC024 | |

## Yeast strains and growth media used

All strains used in this study have been derived from BY4742 (*Brachmann et al., 1998*). The wild-type is AN63, which is derived from BY4742 by making it maltose-prototrophic, reducing its high-petite frequency and finally switching its mating type to *MATa*. Specifically, this strain contains a functional *MAL* regulator allele that was introduced in the locus of *MAL13* on chromosome VII (*Brown et al., 2010*), has been rescued from a frameshift mutation in *SAL1* (*Dimitrov et al., 2009*) and its mating type has been switched to *MATa*. In order to track gene expression, protein fusion constructs with the fluorescent marker *yECitrine* have been constructed for *GAL1*, *MAL11*, *MAL12*, *CIT1*, *MDH1*, *KGD2*, *NDI1*, *SDH2* and *COX6*. In order to see the effect of overexpression, *MAL12*, *MAL11* or both were put under the control of the strong *TDH3*-promoter. The *MAL* gene deletion mutant was constructed by deleting the two complete loci that contain all the *MAL* genes on chromosome II (locus 3) and VII (locus 1). The Bar-Seq experiment was performed using the haploid *MATa* yeast deletion collection, after incorporation of a functional *MAL* regulator and fluorescent protein fusions into each individual mutant using the Synthetic Genetic Array technology (*Tong et al., 2004*).

All experiments were performed at 30°C using rich media. The media that were used were YP (10 g/L yeast extract, and 20 g/L peptone) supplemented with 5% glucose, 10% maltose, 5% galactose or 5% glycerol. A concentrated YP solution is autoclaved, while the concentrated sugar solutions are filter-sterilized. After cooling, they are mixed in the right proportions. In specific experiments, the respiratory inhibitor antimycin A, or the chitin-staining dye Calcofluor White were added to the medium.

## Culturing and growth conditions

To measure the effect of the time grown on glucose on the lag phase length, we pre-grow cultures in the following way. In general, all growth occurs in 150 μL volume in 96-well plates which are sealed with plastic seals to avoid evaporation. In order to control cell density of these cultures, serial dilutions of 50–100 μL into 150 μL are made across the plate before incubation. Incubation lasts maximally 24 hr, after which cultures are transferred to fresh medium. At each transfer, optical density (OD$_{600}$) is measured using a plate reader and only cultures with an OD$_{600}$ <0.05 are selected to be transferred. This OD$_{600}$ threshold corresponds to a cell density <8*10$^6$ cells/mL as measured using a Bio-Rad TC20 Automated Cell Counter. The cell density is controlled in this way throughout the experiment until the final shift from glucose to the alternative sugar, where the cultures experience a lag phase. Only during the specific experiments that were designed to measure the effect of

$OD_{600}$ on the lag time, cultures were allowed to reaches higher densities during the pre-growth (*Figure 1—figure supplements 5* and *6*).

The 2, 4, 6, 8, 10 and 12 hr glucose cultures are grown for two overnights in the pre-glucose sugar (usually maltose or galactose). Next, at different times during the day, cultures are washed twice and transferred to glucose. The 0 hr glucose condition is grown in the same way, but is never transferred to glucose. For the 24 hr glucose condition, cultures are grown for one overnight in the pre-glucose sugar, after which they are washed twice and transferred to glucose for another overnight. The next morning, another dilution is made into fresh glucose to avoid overgrowth. Finally, cultures from all the above conditions with $OD_{600}$ <0.05 are selected at the same time, washed twice and transferred to the post-glucose sugar (usually maltose or galactose). To measure the lag time of these cultures, they are transferred to the Bioscreen C plate reader (population-level measurement) or the microscope for time-lapse imaging (single-cell level measurement). The lag phase under each condition is measured in duplicate or more.

Given the 90 min doubling time in glucose, the 2, 4, 6, 8, 10 and 12 hr of glucose growth corresponds to 1.3, 2.6, 4, 5.3, 6.6 and 8 doublings.

## Population-level lag time measurements using an automated plate reader

To measure the growth dynamics during the lag phase, cultures pre-grown as described above are transferred to the Bioscreen C plate reader. This plate reader allows continuous medium-amplitude shaking and temperature control at 30°C while measuring the $OD_{600}$ every 15 min. Per honeycomb plate, only 20 – 25 wells are filled with culture in such a way that no culture is directly neighboring another. The remaining 75 – 80 well are filled with blank medium. Once all cultures have reached stationary phase, the experiment is stopped.

Population lag times are calculated from the acquired growth curves using R and MS Excel. First, the background $OD_{600}$-signal as measured in the wells containing blank medium is subtracted from all $OD_{600}$-measurements. Second, these $OD_{600}$-values are corrected for the non-linear relation between cell density and optical density using the formula:

$$OD_{600,corrected} = OD_{600,measured} + 0.449 * (OD_{600,measured})2 + 0.191 * (OD_{600,measured})3$$

as derived by *Warringer and Blomberg (2003)*. Third, the discrete derivate of $ln(OD_{600,corrected}600)$ versus time is calculated and this function is smoothed using R's smooth.spline function, with smoothing parameter = 0.40. Fourth, the maximum growth rate ($maxR600$) is calculated by taking the mean of the five largest growth rate values. The time and $OD_{600}$ at which this maximum growth rate occurs (respectively $maxR_{time}600$ and $maxR_{OD}600$) is calculated by taking the mean of the time- and $OD_{600}$-values corresponding to the five largest growth rate values. Fifth, the initial $OD_{600}$ ($OD_{600,initial}600$) is calculated by taking the mean of the first two $OD_{600}$-measurements of each growth curve. Finally, the lag time is calculated using the following formula:

$$Lag_{time} = ln(OD_{600,initial}/exp(ln(maxR_{OD}) - maxR_{time} * maxR))/maxR$$

This calculates the time corresponding to the intersection point between two lines on an $OD_{600}$ versus time graph: (1) a horizontal line crossing the y-axis at $OD_{600,initial}$, and (2) an exponential line tangential to the point of maximum growth, and an exponential parameter equal to the maximum growth rate. These calculations agree with the most common definition of the population lag time after a sudden transition between media (*Swinnen et al., 2004*). The fold increase in population density, as shown in *Figures 1B* and *3I*, is calculated by dividing the $OD_{600,corrected}$-values by the $OD_{600,initial}$.

## Dissolved oxygen measurements

For the measurements during the lag phase, respiratory deficient mutant (cox6::natR), over-active respiratory mutant (pGPD-HAP4) and wild-type strains were pregrown in maltose for two overnights, washed and grown in glucose for 6 hr. The detailed setup of such pre-growth conditions is described above. The cells were washed and transferred to maltose media after this pre-growth. The cultures are then loaded into a 96-well plate with fluorescent oxygen sensor embedded at the bottom of the wells (OxoPlates, PreSens Precison Sensing). To calibrate the oxygen levels, air saturated growth

media was used as the 100% saturated condition, while water containing 10% $Na_2SO_3$ was used as the 0% saturated condition. Wells with only growth media and no inoculation were used as blank controls. The measurements during glucose growth condition are performed similarly except that the cells are pregrown for two overnights in glucose, washed in fresh glucose medium and then transferred to the OxoPlate.

## Single-cell-level lag time measurements using time-lapse microscopy

To measure the growth dynamics during the lag phase at the single-cell-level, cultures pre-grown as described above are sandwiched between an agar pad containing the same medium as used for the final resuspension, and a coverslip (Cerulus et al., 2016; New et al., 2014). The agar pad is wrapped in plastic foil so it does not dry through the experiment. This allows tracking growth and gene expression of hundreds of single cells during the lag phase by periodically (every 15–30 min) taking differential interference contrast (DIC) and fluorescence pictures. The time-lapse movies are acquired automatically by the Metamorph software (version 7.8.0.0; Molecular Device, LLC) in combination with an inverted Nikon Eclipse Ti microscope equipped with a DL-604M-#VP camera (Andor$^{TM}$ technology), and Lambda XL (Sutter Instruments) light source for fluorescent measurements. The microscope is placed in a temperature-controlled incubator (30°C). All images are taken using a 60 × 1.4 NA oil immersion lens, and the image focus is controlled using an image-based Z-plane focusing algorithm. Single-cell lag times are analyzed manually by scoring the time of growth resumption.

For cells with a bud, the time at which the bud increased in size was scored. For cells without a bud, the time of the first morphological change leading to a new bud was scored. Under our experimental set-up, individual cells form microcolonies as they resume growth.

For all the single-cell microscopy experiments, except for the heterokaryon experiment, the microcolony area and gene expression changes (as measured using fluorescent protein fusions) are scored semi-automatically using a pipeline based on MATLAB and CellProfiler scripts (Carpenter et al., 2006). Only microcolonies originating from single cells (with or without a bud) are scored. First, the DIC and fluorescence pictures are aligned correctly. Second, a mask is created using CellProfiler to indicate pixels corresponding to cells (mask value become 1), and pixels which are background (mask value becomes 0). In this way, adjacent pixels corresponding to microcolonies form individual objects. In our pipeline, each image is subjected to a Sobel edge detection algorithm, followed by a thresholding step, and a series of consecutive dilation and erosion steps to generate continuous objects. This procedure is based on a series of image processing steps inspired by Levy et al. (2012); Ziv et al. (2013). Third, microcolony tracking through time is done by (1) a manual selection of objects corresponding to single cells in the initial images, followed by (2) an automatic detection of related objects in the images from the following time points. Tracking of individual colonies is prematurely aborted when the microcolony grows outside of the field of view, or when it collides with another microcolony. Occasional out-of-focus images were excluded from the analysis. Fourth, the microcolony area (number of pixels) and the background-corrected mean fluorescence within each tracked object are calculated. Finally, other qualitative parameters such as the presence of a bud, or the amount of bud scars detected after Calcofluor White staining are scored manually. The induction time of MAL11 and MAL12 fluorescent protein fusions was scored by calculating the time at which the signal increased distinguishably above its initial level. Visualization of the microcolony area and fluorescence changes is done using kymographs, where horizontal lines represent individual microcolonies tracked versus time. The color scale represents either the mean fluorescence or the relative area increase. The individual tracks are sorted based on the pattern of area increase (from an early to a late area increase).

## Time-resolved cross-correlation analysis between microcolony growth rate and protein expression

For microcolonies that resumed growth during the experiment, the growth rate was estimated as a function of time by smoothing the logarithm of that colony's area (i.e., the number of pixels at each time point) over time using a Savitzky-Golay filter (2nd order polynomial; window of length 15) and computing the first order derivative of the filter. Cross-correlation of the growth rate time course with that of the mean fluorescence was performed for each microcolony and the population average

is shown in the cross-correlation plots. Number of cells for each of the tested *yECitrine*-tagged strains: 186 for *MAL11-yECitrine*, 208 for *MAL12-yECitrine*, 151 for *CIT1-yECitrine*, 167 for *MDH1-yECitrine*, 158 for *KGD2-yECitrine*, 157 for *NDI1-yECitrine*, 180 for *SDH2-yECitrine* and 169 for *COX6-yECitrine.*

## Mating experiment

Cells of opposite mating type with genomically-integrated consecutively-expressed fluoresce markers (*MATα pTDH3-mCherry* and *MATa pTDH3-yECitrine*) were pre-grown separately in glucose and maltose media. The conditions of such pre-growth are detailed in 'culturing and growth conditions' section of Materials and Methods. After this pre-growth, the cultures were washed into glucose media. Culture pairs of opposite mating type pre-grown in glucose or maltose media were mixed together in combinations as outlined in *Figure 3—figure supplement 6*. When mixing cultures of opposite mating type, equal number of cells from each culture were mixed together. Mixed cultures of different dilution levels were grown in glucose media for 4 hr. During this period, some cells from the opposite mating types would mate. At the end of glucose growth, cultures were washed into maltose and prepared for time-lapse microscopy on an agar pad with maltose media. The lag time and fluorescence signal were measured for each cell. Depending on the presence of mCherry or yECitrine or both signals, we could determine whether each cell was a *MATα* haploid, or *MATa* haploid or diploid. Cumulative fraction of cells escaping lag phase through time was calculated for each of these groups.

## Heterokaryon experiment

We utilized a karyogamy mutant (*kar1-1*) strain to investigate whether glucose pre-growth time affects the time of *MAL* gene induction due to a cytoplasmic factor or a nuclear factor. The genotype of the two used haploids strains are *MATa MAL12-mCherry kar1-1* and *MATα MAL12-yECitrine KAR1*. After mating of two cells with these genotypes, the resulting diploid cell will have two separate nuclei. As outlined in *Figure 3—figure supplement 7* the two strains were pre-grown in both glucose and maltose separately. Cells from the two strains were not mixed during this pre-growth.

At the end of the pre-growth, the cultures were washed separately to glucose media. Afterwards the glucose pre-grown *MATa* strain was mixed with the maltose pre-grown *MATα* strain with equal cell counts. Similarly, the maltose pre-grown *MATα* strain was mixed with glucose pre-grown *MATa* strain. After making a serial dilution, these mixed cultures along control unmixed cultures and also control mixed cultures with no *kar1-1* mutation were grown in glucose for 4 hr. The controls are not shown here. During this 4 hr of glucose growth, some cells within the mixed cultures of opposite mating type mate. At the end of this glucose growth period, the cultures were washed into maltose and prepared for time-lapse microscopy on an agar pad with maltose media.

For this experiment microscopy images were segmented so that single cells are separated and tracked through time. The fluorescence intensity for mCherry and yECitrine was measured for each cell. Segmentation and tracking of single cells was done using a pipeline based on CellProfiler, Ilastik (*Sommer et al., 2011*) and manual curation. Only the 8-shaped cells (which are the diploids resulted from mating) are considered for the analysis. The induction time of the Mal12-mCherry or MAL12-yECitrine was quantified for each single cell by comparing the average signal within a window of three time points, centered at a given time point to the basal level. The basal level was calculated as the mean signal from time zero to the given time frame. If the difference between these two values was larger than a given threshold, induction time was scored at that given time point. The threshold for this comparison was chosen by inspecting the intensity distribution at the very first time point. For both of the yellow and red channels, each diploid cell initially has one channel at an undetectable level and the other channel at a detectable level. This is because each diploid has risen from two haploids: one haploid with no recent pre-growth in maltose and hence no detectable Mal protein while the other haploid had recent exposure to maltose (4 hr ago) so it has a detectable level of Mal protein. For the yellow channel the mean of this detectable signal level at time point zero is set as the threshold for induction detection. For the red channel due to the lower brightness of mCherry compared to yECitrine, twice the mean of the detectable signal at time point zero is set as the threshold.

Number of cells in *Figure 3—figure supplement 6* upper-left panel is 33 for *MATα*, 38 for *MATa* and 34 for diploids. For the upper-right panel there are 59 *MATα*, 59 for *MATa* and 45 diploid cells. For the bottom-left panel there are 25 *MATα*, 64 for *MATa* and 27 diploid cells. For the bottom-right panel there are 50 *MATα*, 38 for *MATa* and 25 diploid cells.

## Flow cytometry to measure gene expression

Cultures were pre-grown as described above. Samples were taken at different time points by centrifugation and resuspension in 25% glycerol, and frozen at −20°C. All samples were thawed and analyzed on the same day, using the Attune NxT Flow Cytometer (Thermo Fisher Scientific). For each sample, a total of 50,000 single-cell events were acquired. To analyze the yECitrine-signal, excitation at 488 nm and a 530/30 emission detector were used. The analysis was done in R using the flowCore package (*Hahne et al., 2009*). The function 'norm2Filter' with parameter scale = 2 was used to filter out events that did not fit a bivariate normal distribution based on their side-and forward-scatter dimensions. After filtering, the mean fluorescence of each sample was calculated using the arithmetic mean. Changes in gene expression are measured using biological duplicates.

## Construction of a maltose-prototrophic yeast deletion collection

The barcode sequencing experiment was done using the haploid *MATa* yeast deletion collection (*Winzeler et al., 1999*). However, this collection is made using a parental strain that has growth defects on maltose due to the absence of a maltose-responsive MAL-activator (*Brown et al., 2010*; *Pougach et al., 2014*; *Voordeckers et al., 2012*). Therefore, we first constructed a maltose-prototrophic yeast deletion collection, by incorporating a functional *MAL*-regulator into each mutant using the synthetic genetic array (SGA) technology (*Hin Yan Tong and Boone, 2005*; *Xiao, 2006*). The query strain that was used to cross with the deletion collection was the standard query strain Y7092, which was modified to 1) replace the non-functional *MAL13*-activator with the functional *MAL63*-regulator linked to a natamycin resistance marker ($NAT^R$:*MAL63*::*MAL13*), and (2) contain fluorescent protein fusions of *MAL12-yEcitrine* linked to a leucine auxotrophic marker (*LEU2*), and *MAL11-mCherry* linked to a uracil auxotrophic marker. All these modifications occur within a 17 kb region on the subtelomere of chromosome VII. Since this region is flanked by a $NAT^R$ on the centromeric side, and *LEU2* on the telomeric side, this whole region could be selected during SGA by simultaneous selection on natamycin resistance and leucine prototrophy.

In our hands, initially 4722 deletion mutants could be retrieved from our original frozen stock. A total of 616 mutants were lost during the SGA procedure, leading to a maltose-prototrophic collection containing 4128 mutants. In order to generate a pool of deletion mutants that were pre-adapted to maltose growth, these mutants were individually grown for 5 days in 96-well plates containing 150 μL of YP-maltose 2% until most cultures reached stationary phase. All of these cultures were then pooled without accounting for potential differences in cell density between cultures, and aliquots were made of 1 mL.

## Bar-sequencing experiment and analysis

For the Bar-sequencing experiment, cultures were grown in the following way. The first day, one aliquot containing the maltose-adapted gene deletion pool (see above) was inoculated in a 1 L flask containing 500 mL of maltose medium, and incubated overnight. The second day, when the cell density was around $6*10^6$ cells/mL, the culture was split into two. The first part became the '0 h-glucose' condition. A volume of ±50 mL was washed twice with maltose medium, resuspended in a 1 L flask containing 500 mL of maltose medium, and incubated. The second part became the '2,4,6,8,10,12 h-glucose' conditions. A volume of ±200 mL was washed twice with glucose medium and resuspended in the same volume of glucose medium. Different aliquots of this resuspension were used to inoculate different 1 L flasks containing 500 mL of glucose medium, and these flasks were incubated for 2,4,6,8,10 and 12 hr. These aliquots were calculated so that the cell density at the end of glucose growth was always $3$–$4*10^6$ cells/mL. At the end of glucose growth, the cultures were transferred to maltose medium where they experienced a lag phase. This was done in such a way that the initial cell density in maltose was ±$6.25*10^5$ cells/mL. Again, transferring involved two washes and a final resuspension in maltose medium. Finally, these cultures were stopped when they increased to ±$5*10^6$ cells/mL, which corresponds to three population doublings. Cultures were grown in

biological duplicates. At the end of each growth phase (pre-growth maltose, glucose, maltose) a sample was taken. Chloramphenicol was added to the medium to avoid bacterial contamination during the experiment. Cell counts were measured through the experiment using the Bio-Rad TC20 Automated Cell Counter.

DNA was extracted using a Zymolyase-based protocol, and the genomic DNA extracts were equalized based on the cell density of the samples. PCR barcode amplification of the UP and DN barcodes was done using primer sets (UP-tag: U1 and p69; DN-tag: D1 and D2) that generate respectively 168 bp and 190 bp fragments containing these barcodes at one of their ends. Ex-Taq polymerase (TaKaRa) was used using standard cycling conditions, including 30 s of elongation time and an annealing temperature of 55°C. A limited number of amplification cycles (16) was done to avoid issues associated with over-amplification. Finally, PCR product from the UP and DN reaction were pooled per sample and were sequenced using Illumina NextSeq 500. Analysis of the sequencing reads was done by blasting them to in-silico generated sequences corresponding to the different gene deletion mutants, allowing for 2 mismatches and three gaps. The frequency of each mutant in a sample was calculated by dividing its count by the total count of matched barcodes.

In order to determine growth rates and the lag time, the frequencies were multiplied by the total population growth of the culture during the experiment. This transforms relative growth (change in frequency) to absolute growth (change in absolute numbers), and allow calculation of growth rate and lag, when the following assumptions are made. To calculate the adapted maltose growth rate ($GR_{mal}$), we assume exponential growth during the final round of maltose growth of the '0 h-glucose' condition. This growth rate was calculated using the formula:

$$GR_{mal} = ln\big(cell\_count_{final}/cell\_count_{initial}\big)/\big(time_{final}-time_{initial}\big)$$

with $cell\_count$ and $time$ representing the absolute cell number and time at the start (initial) and end (final) of the maltose growth period. To calculate the adapted glucose growth rate, we assume exponential growth between 4 and 12 h after the transition to glucose. This growth rate is calculated by extracting the slope ($\beta_1$) from the linear model:

$$ln\big(cell\_count_{final,i}/cell\_count_{initial,i}\big) = \beta_0 + \beta_1\big(time_{final,i}-time_{initial,i}\big) + \varepsilon_i$$

with $cell\_count$ and $time$ representing the absolute cell number and time at the start (initial) and end (final) of the glucose growth period for each $i$-th condition ('4,6,8,10,12 h-glucose' conditions). To calculate the lag time, we assume that cells experience a period of no growth, after which they immediately resume growth at the adapted maltose growth rate. The formula to calculate the lag time is:

$$Lag_{time} = ln\big(cell\_count_{initial}/exp\big[ln\langle cell\_count_{final}\rangle - GR_{mal}*\langle time_{final}-time_{initial}\rangle\big]\big)/GR_{mal}$$

with $cell\_count$ and $time$ representing the absolute cell number and time at the start (initial) and end (final) of the final maltose period for each condition ('2,4,6,8,10,12 h-glucose' conditions).

All calculations were performed separately on the data from the two biological replicates A and B, and their technical replicates UP and DN tags. The calculation based on any of these four replicate measures for a specific mutant was discarded when it had 1) a count = 0 of any sample, or (2) a count <30 at the end of any of the glucose conditions. When both technical replicates (UP and DN) were valid, the lag time and growth rates were averaged per biological replicate. Otherwise, only the valid technical replicate was used. Finally, the mean and SD of the biological replicates was calculated, and are used in *Figure 4A*, *Figure 4—figure supplement 4* and *Figure 5—figure supplement 3*.

Different classes of deletion mutants with altered HDB dynamics were obtained: (1) consistently shorter lag, (2) consistently longer lag, (3) long-then-short lag, and (4) short-then-long lag. This was based on their lag times in the 2–12 hr glucose conditions relative to the other mutants:

| | | Time in glucose(h) | | | | | |
|---|---|---|---|---|---|---|---|
| | | 2 | 4 | 6 | 8 | 10 | 12 |
| Altered hysteresis class | Short | <30% | <20% | <15% | <10% | <10% | <10% |
| | Long | >92.5% | >90% | >90% | >90% | >90% | >90% |
| | Long-then-short | >85% | >50% | | | <50% | <30% |
| | Short-then-long | <15% | <30% | | | | >50% |

These classes contain respectively 195, 190, 75 and 72 mutants. GO enrichment analysis was performed on these gene lists using the GO Term Finder tool (*Boyle et al., 2004*), using a Holm-Bonferroni-corrected p-value<0.05. The interactions between these genes were investigated using the PheNetic interaction network (*De Maeyer et al., 2015*). This network is based on protein-protein, protein-DNA and phosphorylation interactions. The subnetworks created by selecting the different classes of deletion mutants are visualized using Cytoscape (*Shannon et al., 2003*).

## RNA-sequencing experiment and analysis

Single colonies of AN148 (AN63 containing *MAL11-yECitrine* and *MAL12-mCherry*) were inoculated in maltose and were grown for three overnights in dilute conditions. Next, these cultures were washed into glucose and different volumes were inoculated so that after different times in glucose (between 0 and 12 hr) they reached a cell count of $<2.5*10^6$ cells/ml. These glucose cultures were then washed into maltose where they experience a lag phase. Chloramphenicol was added to the medium to avoid bacterial contamination during the experiment. Two biological replicates were used in parallel. Samples for mRNA sequencing were taken throughout the experiment and immediately mixed with equal amounts (25 mL) of ice-cold DEPC-treated water. This mix was centrifuged, washed with in 1 mL of ice-cold DEPC-treated water and transferred to an Eppendorf tube, centrifuged again and frozen after removal of the supernatant. Whole RNA was extracted using the MasterPure^TM Yeast RNA Purification Kit (Epicentre) and was sequenced using Illumina HiSeq2500. Sequencing reads were mapped to the reference genome using tophat2 (*Kim et al., 2013*). The resulting bam files were converted to sam format using samtools (*1000 Genome Project Data Processing Subgroup et al., 2009*). The number of mapped reads covering each gene were counted using HTSeq (*Anders et al., 2015*).

The R package DESeq2 was used to detect differentially expressed genes across the different samples (*Love et al., 2014*). For this analysis, the sample taken during pre-growth in maltose was always used as a reference, and the cut-off for significance was an adjusted p-value<0.01. Genes that were detected in less than 15 out of 29 conditions were not included in the analysis. This analysis generated a list of significantly up- and down-regulated genes per sample, and their associated $log_2$ expression change relative to the reference condition. These gene lists were analyzed using GO enrichment for 'biological processes' using the GO Term Finder tool (*Boyle et al., 2004*). GO categories with a Holm-Bonferroni-corrected p-value<0.05 were considered to be significantly enriched within the lists of up- and down-regulated genes. Since this returned many GO categories, we focused on a smaller number of relevant categories using two selection steps. First, GO categories that were significantly enriched in either up- and down-regulated genes in less than 4 out of the 29 samples were discarded, leading to a reduction from 486 to 285 GO terms. Second, redundant GO terms were removed using REVIGO filtering out categories with a similarity >0.5 (*Supek et al., 2011*), reducing the total number of 83 GO terms.

The mean and slope of the expression change during glucose (relative to the maltose pre-growth condition) were calculated based on the samples taken in glucose, and excluding the initial reference conditions. The mean was calculated from by taking the arithmetic mean of the $log_2$ expression changes, while the slope ($\beta_1 2$) was calculated using a simple linear regression model of $log_2$ expression change versus the time grown in glucose:

$$log_2(expression_i) = \beta_0 + \beta_1 (time_{glucose,i}) + \varepsilon_i$$

with *expression* being the expression level in glucose relative to the reference maltose conditions, $time_{glucose,i}$ representing the time in glucose, and '*i*' representing the different glucose conditions.

## Acknowledgments

We thank the GeneCore Sequencing Facility (EMBL, http://www.genecore.embl.de) for sequencing of the RNA-Seq experiment. We thank VIB Nucleomics Core for sequencing of the Bar-Seq experiment. This work was supported by FWO PhD grant for BC JMJP received support from the SULSA Postdoctoral Exchange Scheme. Research in the laboratory of KJV is supported by VIB, AB-InBev-Baillet Latour Fund, FWO, VLAIO, and European Research Council (ERC) Consolidator Grant CoG682009. KJV and PSS acknowledge funding from the Human Frontier Science Program (HFSP) grant 246 RGP0050/2013.

## Additional information

### Funding

| Funder | Grant reference number | Author |
|---|---|---|
| Fonds Wetenschappelijk Onderzoek | | Bram Cerulus<br>Lieselotte Vermeersch |
| Vlaams Instituut voor Biotechnologie | | Kevin J Verstrepen |
| European Research Council | CoG682009 | Bram Cerulus<br>Abbas Jariani<br>Gemma Perez-Samper<br>Kevin J Verstrepen |
| AB-InBev-Baillet Latour Fund | | Kevin J Verstrepen |
| Human Frontier Science Program | 246 RGP0050/2013 | Abbas Jariani<br>Peter S Swain<br>Kevin J Verstrepen |
| SULSA Postdoctoral Exchange Scheme | | Julian M J Pietsch |

The funders had no role in study design, data collection and interpretation, or the decision to submit the work for publication.

### Author contributions

Bram Cerulus, Conceptualization, Data curation, Software, Formal analysis, Validation, Investigation, Visualization, Methodology, Writing—original draft, Project administration, Writing—review and editing; Abbas Jariani, Conceptualization, Data curation, Software, Formal analysis, Validation, Investigation, Visualization, Methodology, Writing—review and editing; Gemma Perez-Samper, Conceptualization, Investigation, Methodology, Writing—review and editing; Lieselotte Vermeersch, Investigation, Writing—review and editing; Julian MJ Pietsch, Software, Investigation, Methodology, Writing—review and editing; Matthew M Crane, Methodology, Contributed substantially to acquisition of data by contributing to development of the microscopy techniques used in this manuscript; Aaron M New, Conceptualization, Software, Investigation, Methodology; Brigida Gallone, Software, Contributed substantially to analysis and interpretation of data for the BAR-Seq experiment; Miguel Roncoroni, Investigation, Contributed substantially to acquisition of data and doing the experiments; Maria C Dzialo, Writing—review and editing, Contributed substantially to revising the article critically for important intellectual content; Sander K Govers, Investigation, Methodology; Jhana O Hendrickx, Eva Galle, Maarten Coomans, Pieter Berden, Sara Verbandt, Investigation, Contributed substantially to acquisition of data, doing the experiments and construction of the strains; Peter S Swain, Conceptualization, Funding acquisition, Writing—review and editing; Kevin J Verstrepen, Conceptualization, Resources, Supervision, Funding acquisition, Writing—original draft, Project administration, Writing—review and editing

### Author ORCIDs

Abbas Jariani [iD] http://orcid.org/0000-0003-2715-933X
Lieselotte Vermeersch [iD] http://orcid.org/0000-0001-5789-2220

Julian MJ Pietsch (iD) https://orcid.org/0000-0002-9992-2384
Sander K Govers (iD) http://orcid.org/0000-0003-0260-4054
Peter S Swain (iD) http://orcid.org/0000-0001-7489-8587
Kevin J Verstrepen (iD) http://orcid.org/0000-0002-3077-6219

**Decision letter and Author response**
Decision letter https://doi.org/10.7554/eLife.39234.047
Author response https://doi.org/10.7554/eLife.39234.048

## Additional files

### Supplementary files

• Supplementary file 1: Analysis of Bar-Seq data. This file contains the data and procedure for obtaining lag times in the Bar-Seq experiment starting from read counts. The procedure is detailed in the Materials and Methods section.
DOI: https://doi.org/10.7554/eLife.39234.038

• Supplementary File 2 RNA-Seq count data. This file contains expression values (fpm) for RNA-Seq samples. Each column represents one sample point. At each sampling point two biological replicates are sampled. These replicates are named 'A' or 'B'.
DOI: https://doi.org/10.7554/eLife.39234.039

• Supplementary File 3 Annotation of the sample numbers in RNA-Seq count data. This file contains the description of time point, media and pre-growth conditions for each of the sample numbers in RNA-Seq count data from *Supplementary File 1*.
DOI: https://doi.org/10.7554/eLife.39234.040

• Transparent reporting form
DOI: https://doi.org/10.7554/eLife.39234.041

### Data availability

The RNA-seq and BAR-seq data-sets are deposited in GEO. The GEO accession number of BAR-Seq and RNA-Seq data are GSE116505 and GSE116246 respectively.

The following datasets were generated:

| Author(s) | Year | Dataset title | Dataset URL | Database and Identifier |
|---|---|---|---|---|
| Cerulus B, Jariani A | 2018 | BAR-Seq to study history-dependent behavior | https://www.ncbi.nlm.nih.gov/geo/query/acc.cgi?acc=GSE116505 | NCBI Gene Expression Omnibus, GSE116505 |
| Jariani A, Cerulus B | 2018 | Transition between fermentation and respiration determines history-dependent behavior in fluctuating carbon sources | https://www.ncbi.nlm.nih.gov/geo/query/acc.cgi?acc=GSE116246 | NCBI Gene Expression Omnibus, GSE116246 |

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
