## [Decision Letter]

Thank you for submitting your article "Transition between fermentation and respiration determines history-dependent behavior in fluctuating carbon sources" for consideration by *eLife*. Your article has been reviewed by Naama Barkai as the Reviewing and Senior Editor, a Reviewing Editor, and three reviewers. The following individual involved in review of your submission has agreed to reveal his identity: Florian Bauer (Reviewer #3).

The reviewers have discussed the reviews with one another and the Reviewing Editor has drafted this decision to help you prepare a revised submission.

As you will see, the reviewers all recommended publications, but have raised some important points, please address them in full. Of particular note are the controls requested by reviewer 3, and the need to examine additional strains, as requested by reviewer 2.

*Reviewer #1:*

The ability of *Saccharomyces cerevisiae* to respond more quickly to galactose given a previous exposure to galactose was an interesting and 'tractable' example of history dependent behavior (HDB). Before Cerulus et al., multiple papers had examined this specific phenomenon and had focused on either persistent proteins or chromatin state as being the critical mechanism for the "cellular memory". In this paper, the authors show the surprising result that sugar HDB has a component that is not specific to the specific sugar, i.e. maltose can give galactose HDB and vice versa. The authors then track down the mechanism and give several lines of correlative evidence (microarray and bar-seq) to suggest metabolic state can play a critical role. They then tighten the correlation through single cell analysis of gene expression and perturbation with the metabolic state. I can come up with experiments that I might have done if my lab had made this observation or a slightly different focus, e.g. determine more quantitively the contribution and kinetics of protein persistence and metabolic state, but I believe the work performed is of sufficient quality and interest to warrant publication. Because one could argue that we already know that metabolic state can affect sugar response, in retrospect this result could to some seem trivial. But, typically it was thought that the switch from secondary to preferred carbon source was rapid and it was only the reverse directly where there was a strong pregrowth effect. No one had clearly established the role of metabolic state in the HDB and hence I think this paper both leads to more clarity in the field and will spark more careful and potentially critical analysis of how cells transition from respiration to fermentation.

Reviewer #2:

In this manuscript Cerulus et al., investigated the nature and molecular cause of the growth lag that is observed when yeast cells are shifted from a fermentable to a non-fermentable carbon source. They report that the lag depends on the exact pre-growth regimen, whether the cells have already before experienced a non-fermentable carbon source, and how long they have been growing on the fermentable carbon source meanwhile. They employ population level and single cell studies to characterize population diversity and growth behaviour. In the first part of the paper they explore this phenomenon using different types of growth regimen and they conduct several experiments to follow up on the question whether resumption of growth is correlated with the expression of the non-fermentable sugar specific genes (MAL and GAL genes). While they report that higher levels of MAL genes are prompting growth, they argue, also using gene expression tracking in single cells, that the lag phase is not determined by the time it requires to express these sugar specific genes, in individual cells or on the level of populations, but that other factors are limiting.

They subsequently conducted two types of screens to identify factors and genes related to the length of the lag phase and how it is impacted by the pre-growth regimen, i.e. using the barcoded yeast deletion collection, and mRNA abundance measurements.

Taken together this analysis provides a main insight that led to the hypothesis that the deterministic factor for the lag phase is the presence of respiratory components at the time point of the fermentation->respiration switch. They convincingly argue that these proteins (or the organelle, where they reside) becomes diluted upon a shift to a non-fermentable carbon source and only cells that have lost this 'equipment' are prone to a strong lag. They provide experiments in support of this hypothesis, i.e. by showing that the lag phase correlates with the amount/activity of a transcription factor involved in the regulation of respiratory genes.

Altogether I found the work very well executed, convincing and relatively easy to follow, despite the complexity of the subject. While the work clearly requires "yeast" knowledge for full appreciation, it still serves as a paradigm for studies in other cells and organisms where large metabolic shifts matter.

As a non-insider to this exact field I found the first part of the introduction very reference-sparse and it was hard to for me judge the knowledge-base of the initial statements, in particular the first paragraph that introduces 'HDB'.

Table with strain names and exact genotypes of strains and ancestry should be included.

Plasmid constructions and a plasmid list should be included.

Any paper dealing with complex growth regimen involving different media should carefully list the source of all media components. The same is true for sugars, here also the exact quality needs to be detailed, since sugars are sometimes cross-contaminated with each other, and traces of glucose in e.g. galactose can ruin a growth experiment if the inoculum is too diluted (and it takes ages for the cells to eat up the glucose before they can respond to the galactose).

Screen – screens need independent validations, and at last a few hits should be tested individually.

Strain background: The study is based on work exclusively done with the BY4741 strain, although they used a variant that has been cured for the mitochondria related defect. Still, it would be nice to have some overview about HDB penetrance in other backgrounds, diverse other lab strains and wild type strains that have not undergone any laboratory evolution. I find this an important point, because the Gal promoter is (was?) the workhorse of a regulatable promoter in yeast. While many labs use pre-growth on raffinose – Galactose addition then yields a synchronous and rapid induction of the Gal (but not Mal) promoter- other labs are not aware of this and the literature is full of Glc -> Gal switches to induce the Gal promoter. Would be nice to know how bad the lag can get in different strains. Does not need experiments, if there is some data or knowledge around?

*Reviewer #3:*

The manuscript describes a well designed, holistic and thorough examination of history-dependent behaviour (HDB) of yeast cells in fluctuating carbon sources. The authors proceed logically and use state-of-the-art methodologies at both population (culture behaviour, RNAseq), individual cell (bar-code deletion mutants, single cell microscopy) and selected individual gene and protein levels to characterize the molecular mechanism underlying HDB when cells are transferred from one carbon source to another. Overall, they generate a very impressive data set.

The findings are of general interest and suggest that previously reported mechanisms, in particular the persistence of certain proteins required for carbon source utilization after transfer, may not primarily account for HBD, but rather that the general physiological state of the cells, and in particular respiration-related factors, are responsible for the phenomenon.

I have some general suggestions for further improvement of the manuscript:

1) There is one important control missing: A transfer Glu-Glu-Glu.

The different periods of times spend on glucose in the experimental set-up will indeed not only impact on the carbon source, but all other nutrients (and in this context oxygen is a nutrient as well). It is essential to show that a transfer back to Glu, instead of to Mal and Gal, will or will not result in (proportionally) similarly extended lag phases. Every transfer involves exposure to a fresh medium with 100% dissolved oxygen from media which have been differentially depleted of other nutrients as a function of time. Many of these nutrients could in principle cause delays in the lag phase by requiring some re-adjustment (if such nutrients had been exhausted or limited in the previous culture).

2) The authors present all the data in terms of time spend in the glucose medium, but in the text sometimes refer to generations (for example subsection “HDB does not depend on Mal or Gal protein inheritance”). Since "dilution" of proteins is (one of) the main hypothesis that the authors are testing, this information (number of generations) should be included in the data sets.

3) Respiration and link to a different form a lag phase, the diauxic shift, has been studied in yeast physiology. While of a somewhat different nature, such studies have highlighted significant strain-specific differences, and transcriptomic data sets are available. The discussion would benefit from highlighting similarities and differences between the data here and changes during the diauxic shift.

4) The authors hypothesize that the cells enter an energy deficient state after transfer (Discussion section), a reasonable hypothesis. However, I am less convinced by the link to "growth on glucose" as the primary driver of the system. The problem lies in the fact that the authors are in a self-anaerobic system (as the dissolved oxygen data show). The question therefore is whether it is "growth on glucose" or lack of oxygen that drives the system. For this purpose, it would be necessary to repeat the same experiment in a fully oxygenated environment. Considering the quality and overall significant contributions of the manuscript, I would only ask the authors to discuss the potential role of oxygen and of oxygen supply (and not only of glucose) on the system. There are data that suggest that the primary driver of adaptation for fermentative growth is oxygen availability, and not carbon source.

---

## [Author Response]

Reviewer #2:In this manuscript Cerulus et al., investigated the nature and molecular cause of the growth lag that is observed when yeast cells are shifted from a fermentable to a non-fermentable carbon source. They report that the lag depends on the exact pre-growth regimen, whether the cells have already before experienced a non-fermentable carbon source, and how long they have been growing on the fermentable carbon source meanwhile. They employ population level and single cell studies to characterize population diversity and growth behaviour. In the first part of the paper they explore this phenomenon using different types of growth regimen and they conduct several experiments to follow up on the question whether resumption of growth is correlated with the expression of the non-fermentable sugar specific genes (MAL and GAL genes). While they report that higher levels of MAL genes are prompting growth, they argue, also using gene expression tracking in single cells, that the lag phase is not determined by the time it requires to express these sugar specific genes, in individual cells or on the level of populations, but that other factors are limiting.They subsequently conducted two types of screens to identify factors and genes related to the length of the lag phase and how it is impacted by the pre-growth regimen, i.e. using the barcoded yeast deletion collection, and mRNA abundance measurements.Taken together this analysis provides a main insight that led to the hypothesis that the deterministic factor for the lag phase is the presence of respiratory components at the time point of the fermentation->respiration switch. They convincingly argue that these proteins (or the organelle, where they reside) becomes diluted upon a shift to a non-fermentable carbon source and only cells that have lost this 'equipment' are prone to a strong lag. They provide experiments in support of this hypothesis, i.e. by showing that the lag phase correlates with the amount/activity of a transcription factor involved in the regulation of respiratory genes.Altogether I found the work very well executed, convincing and relatively easy to follow, despite the complexity of the subject. While the work clearly requires "yeast" knowledge for full appreciation, it still serves as a paradigm for studies in other cells and organisms where large metabolic shifts matter.As a non-insider to this exact field I found the first part of the introduction very reference-sparse and it was hard to for me judge the knowledge-base of the initial statements, in particular the first paragraph that introduces 'HDB'.

We now have now added more references to the statements in the first paragraph of the Introduction.

Table with strain names and exact genotypes of strains and ancestry should be included.Plasmid constructions and a plasmid list should be included.Any paper dealing with complex growth regimen involving different media should carefully list the source of all media components. The same is true for sugars, here also the exact quality needs to be detailed, since sugars are sometimes cross-contaminated with each other, and traces of glucose in e.g. galactose can ruin a growth experiment if the inoculum is too diluted (and it takes ages for the cells to eat up the glucose before they can respond to the galactose).

We now have included a Key Resources Table with the detailed information of strains, plasmids and the media components used.

Screen – screens need independent validations, and at last a few hits should be tested individually.

Regarding the RNA-Seq, we validate the screen by measuring fluorescent protein fusions and show that the trends that we see in RNA-Seq also hold true there for protein fusions (Figure 5—figure supplement 4).

We have now added validation of the BAR-Seq by making new strains and testing three of the hits (Figure 4—figure supplement 6).

Strain background: The study is based on work exclusively done with the BY4741 strain, although they used a variant that has been cured for the mitochondria related defect. Still, it would be nice to have some overview about HDB penetrance in other backgrounds, diverse other lab strains and wild type strains that have not undergone any laboratory evolution. I find this an important point, because the Gal promoter is (was?) the workhorse of a regulatable promoter in yeast. While many labs use pre-growth on raffinose – Galactose addition then yields a synchronous and rapid induction of the Gal (but not Mal) promoter- other labs are not aware of this and the literature is full of Glc -> Gal switches to induce the Gal promoter. Would be nice to know how bad the lag can get in different strains. Does not need experiments, if there is some data or knowledge around?

We have tested two non-laboratory strains from Cubillos et al., 2009 (Figure 1—figure supplement 4). The HDB is present in both of these feral strains. One feral strain closely mimics the behavior of the original strain, the other one shows a less dramatic change in lag duration. In fact, for a follow-up study, we have now started a QTL analysis to identify alleles that affect the HDB (to be published in a next paper).

Reviewer #3:The manuscript describes a well designed, holistic and thorough examination of history-dependent behaviour (HDB) of yeast cells in fluctuating carbon sources. The authors proceed logically and use state-of-the-art methodologies at both population (culture behaviour, RNAseq), individual cell (bar-code deletion mutants, single cell microscopy) and selected individual gene and protein levels to characterize the molecular mechanism underlying HDB when cells are transferred from one carbon source to another. Overall, they generate a very impressive data set.The findings are of general interest and suggest that previously reported mechanisms, in particular the persistence of certain proteins required for carbon source utilization after transfer, may not primarily account for HBD, but rather that the general physiological state of the cells, and in particular respiration-related factors, are responsible for the phenomenon.I have some general suggestions for further improvement of the manuscript:1) There is one important control missing: A transfer Glu-Glu-Glu.The different periods of times spend on glucose in the experimental set-up will indeed not only impact on the carbon source, but all other nutrients (and in this context oxygen is a nutrient as well). It is essential to show that a transfer back to Glu, instead of to Mal and Gal, will or will not result in (proportionally) similarly extended lag phases. Every transfer involves exposure to a fresh medium with 100% dissolved oxygen from media which have been differentially depleted of other nutrients as a function of time. Many of these nutrients could in principle cause delays in the lag phase by requiring some re-adjustment (if such nutrients had been exhausted or limited in the previous culture).

We have now added the requested glucose→ glucose→ glucose control (Figure 1—figure supplement 3), the cells do not show a lag phase.

There is no distinguishable difference between different conditions. Such absence of effective nutrient depletion might be explained by the fact that we keep the cells in dilute conditions throughout the growth.

2) The authors present all the data in terms of time spend in the glucose medium, but in the text sometimes refer to generations (for example subsection “HDB does not depend on Mal or Gal protein inheritance”). Since "dilution" of proteins is (one of) the main hypothesis that the authors are testing, this information (number of generations) should be included in the data sets.

We now have added this information to the Materials and methods section.

3) Respiration and link to a different form a lag phase, the diauxic shift, has been studied in yeast physiology. While of a somewhat different nature, such studies have highlighted significant strain-specific differences, and transcriptomic data sets are available. The discussion would benefit from highlighting similarities and differences between the data here and changes during the diauxic shift.

We have now discussed this in the Discussion section.

4) The authors hypothesize that the cells enter an energy deficient state after transfer (Discussion section), a reasonable hypothesis. However, I am less convinced by the link to "growth on glucose" as the primary driver of the system. The problem lies in the fact that the authors are in a self-anaerobic system (as the dissolved oxygen data show). The question therefore is whether it is "growth on glucose" or lack of oxygen that drives the system. For this purpose, it would be necessary to repeat the same experiment in a fully oxygenated environment. Considering the quality and overall significant contributions of the manuscript, I would only ask the authors to discuss the potential role of oxygen and of oxygen supply (and not only of glucose) on the system. There are data that suggest that the primary driver of adaptation for fermentative growth is oxygen availability, and not carbon source.

This could have been a possibility if the cultures were grown to higher cell densities, however in these experiments the cells are kept at OD_600_ below 0.05 and as shown in Figure 6—figure supplement 3 (top panel) in these conditions, the oxygen is above 85-90% of the saturation control.